# Demystifying Mergeability:
# Interpretable Properties to Predict Model Merging Success

**Luca Zhou** [1]   **Bo Zhao** [2]   **Rose Yu** [2]   **Emanuele Rodolà** [1] [3]

## Abstract

Model merging combines knowledge from separately fine-tuned models, yet the factors driving its success remain poorly understood. While recent work treats mergeability as an intrinsic property of the models, we show with an architecture-agnostic framework that it fundamentally depends on both the merging method and the partner tasks. Using L1-regularized linear optimization over a set of interpretable pairwise metrics (e.g., gradient $L_2$ distance), we uncover properties correlating with post-merge normalized accuracy across five merging methods. We find that the drivers of merge success vary across architectures and merging methods, overall with only a moderate agreement ($64.0\%$ average top-5 metric overlap; $79.3\%$ sign agreement). Crucially, however, *gradient alignment* metrics consistently emerge as the most fundamental signals of mergeability. These findings provide a diagnostic foundation for understanding mergeability and motivate future merge-aware fine-tuning strategies.

## 1. Introduction

Model merging (Wortsman et al., 2022) has emerged as a practical approach for combining knowledge across multiple fine-tuned neural networks without retraining. Given multiple task-specific fine-tuned models, merging aims to produce a single model that performs well on multiple tasks simultaneously. This is particularly valuable in settings where computational resources are limited or where a unified model is preferred over task-specific checkpoints.

Despite its practical appeal, model merging remains highly unpredictable. Some model pairs merge seamlessly, yielding performance close to the average of their individual task accuracies. Others suffer catastrophic performance degradation due to task interference. This inconsistency motivates a fundamental question: *what determines the mergeability between models?* Recent work (Rahamim et al., 2026) has proposed that mergeability is an intrinsic property of individual models: if a model is mergeable, it remains so regardless of its merge partners. Notably, their analysis is restricted to the LoRA merging setting and is primarily based on KNOTS (Stoica et al., 2025) only. Under this view, the challenge reduces to identifying and certifying models with high mergeability potential. Complementary to this framing, we note that this perspective implicitly aggregates multiple sources of variation, including the properties of both merged models and the choice of merge algorithm, into a single scalar notion of mergeability. Consequently, it remains unclear how mergeability patterns may change across different merging methods or partners.

We test this hypothesis using a flexible, architecture- and method-agnostic framework that applies linear optimization over extensible sets of interpretable pairwise metrics. This approach serves as a scientific probe to isolate the geometric and functional properties associated with model mergeability, revealing how these requirements shift across merging mechanisms (see Figure 1). We emphasize that our goal is not to maximize predictive performance, but rather to *unveil the underlying signals of successful merging*.

Applied to five representative merge methods, namely Task Arithmetic (TA) (Ilharco et al., 2023), Weight Averaging (WA) (Wortsman et al., 2022), Task Singular Vector (TSV) merging (Gargiulo et al., 2025), TIES (Yadav et al., 2023), and DARE (Yu et al., 2023), using 28 pairwise metrics spanning weight-space, activation-, and gradient-based measures, our analysis reveals that there is **no consensus** on "success fingerprint" among all the methods. The substantial variation in optimized coefficients, with top metric overlap and sign agreement respectively reaching a minimum of 40% and 57%, demonstrates that mergeability is not a static and intrinsic model property, but a dynamic, method-dependent relationship. Notably, the same metric can even exert opposite effects across different merging methods.

Our framework delivers three core advantages. First, **predictive structure**: by optimizing for generalization under

---

[1]Sapienza University of Rome [2]UC San Diego [3]Paradigma. Correspondence to: Luca Zhou <luca.zhou@uniroma1.it>.

*Proceedings of the $43^{rd}$ International Conference on Machine Learning*, Seoul, South Korea. PMLR 306, 2026. Copyright 2026 by the author(s).

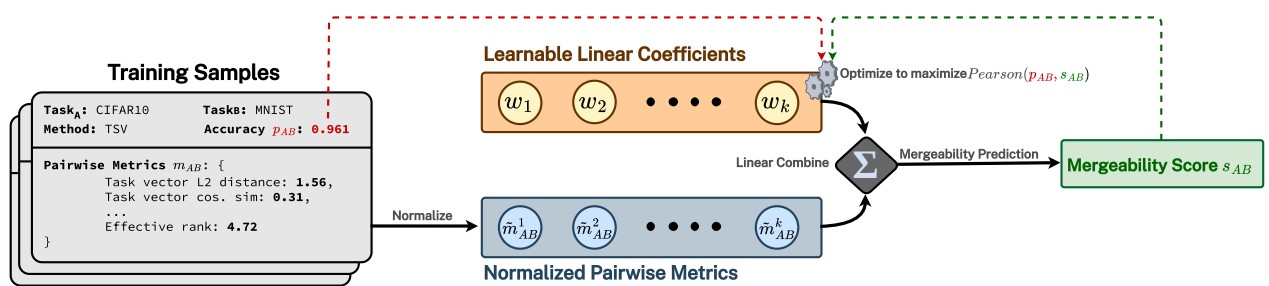

*Figure 1.* Linear optimization procedure of coefficients to maximize the Pearson correlation of the predicted mergeability score with the true post-merge normalized accuracy. Repeated for all merging methods and task pairs, yielding per-method optimal coefficients to predict mergeability. Metrics and true post-merge accuracies are all pre-computed offline; only the linear coefficients are optimized.

leave-one-task-out evaluation, the framework makes testable predictions about merging outcomes on unseen tasks and model pairs, rather than providing post-hoc explanations alone. Second, **full interpretability**: unlike black-box predictors (Bolton et al., 2026), transparent coefficients reveal which model properties are predictive of success for each merging method. Third, **flexibility**: the same optimization can be applied to new merge algorithms or architectures, enabling practitioners to identify and target the properties most associated with successful merging[1]. Finally, **actionability**: we demonstrate the practical utility of our framework via a toy experiment, where insights derived from our analysis directly inform a fine-tuning strategy that improves mergeability. Our contributions are summarized as:

1. **Diagnosing the nature of mergeability**: We introduce an interpretable architecture-agnostic approach that uses linear optimization as a diagnostic tool to identify pairwise model properties that correlate with merging success across merging methods, moving beyond the "intrinsic mergeability" hypothesis.

2. **Discovery of success fingerprints**: We demonstrate that while mergeability requirements are highly method-dependent, some properties, such as gradient alignment, serve as foundational, method-agnostic signals of universal model mergeability.

3. **Actionable and extensible framework**: Linear models achieve meaningful prediction (Pearson $r \in [0.45, 0.70]$, $p \ll 0.001$) with full transparency vs. black-box approaches. Practitioners can apply our framework to any new method/architecture to identify key properties to encourage during fine-tuning for enhanced mergeability, as we show in a toy experiment.

## 2. Related Work

**Model merging and task vectors.** Model merging aims to combine multiple models into one without additional joint

training and adaptation. The purpose of model merging can be two-fold: i) merging models trained on the same task can enhance robustness (Crisostomi et al., 2024), while ii) merging models trained on different tasks enables multi-task capabilities (Li et al., 2023). This work focuses on the latter setting. Prior work has explored simple parameter-space operators such as linear interpolation (Wortsman et al., 2022; Frankle et al., 2020), task vector arithmetic (Ilharco et al., 2023), as well as more structured variants that account for layer-wise geometry and task interference (Yadav et al., 2023; Davari & Belilovsky, 2024; Deep et al., 2024; Wang et al., 2024). Task-vector-based methods view fine-tuning as learning a displacement in weight space and study how these displacements can be composed to transfer or combine capabilities across tasks. More recent approaches refine this view by connecting task vectors to gradients (Zhou et al., 2025a; Daheim et al., 2024), and exploit the low-rank structure of weight matrices (Gargiulo et al., 2025; Marczak et al., 2025). For example, methods based on Task Singular Vectors (TSVs) decompose layer-wise task updates and explicitly reduce inter-task interference before merging, achieving stronger performance than naive arithmetic combinations. These works highlight that different merge operators implicitly rely on different geometric assumptions about how task updates interact.

**Intrinsic mergeability.** A recent line of work seeks to explain *why* some models are more easily merged than others by introducing explicit notions of "mergeability". It has been shown that overtraining task-specific models hurts mergeability due to over-specialization (Horoi et al., 2025; Zhou et al., 2025b). More closely related to this work, Rahamim et al. (2026) define a mergeability score based on repeated merge-and-evaluate trials with random partners, suggesting that mergeability can be treated as an intrinsic property, primarily governed by the base model's prior knowledge. In this view, a highly mergeable model is expected to remain so across different merge partners.

In this work, we adopt a complementary perspective. Rather than treating mergeability as a partner-independent scalar,

---

[1]Our codebase is available at https://github.com/Rose-STL-Lab/demystifying-mergeability.

*Table 1.* Summary of mergeability metrics by category.

| Category | Metrics | Count |
|---|---|---|
| Task Vector | cos sim, $L_2$ dist, dot, angle, magnitude ratio | 5 |
| Effective Rank | eff rank, stable rank, spectral gap, SV ratio | 7 |
| Subspace Overlap | SV overlap, subspace overlap, interaction | 6 |
| Activation-Based | $L_2$ dist, cosine sim, magnitude ratio, dot | 4 |
| Gradient-Based | encoder/input grad. $L_2$ dist., cos sim, dot | 6 |
| **Total** | | **28** |

we examine how the properties predictive of merging success depend on the choice of merge algorithm and vary across merge partners. Additionally, instead of inferring mergeability solely from black-box evaluation trials, we analyze a set of interpretable pairwise metrics to characterize the factors associated with successful merging. This enables a more granular analysis that moves beyond identifying *whether* a model is mergeable to understanding *under which conditions* and *by which signals* mergeability arises.

**Predicting merge outcomes from similarity signals.** Another strand of work explores using similarity signals to predict merge outcomes (Yang et al., 2023; Matena & Raffel, 2022), with works like Bolton et al. (2026) employing black-box models to optimize operator selection and merge orders. While these approaches demonstrate the predictive utility of similarity metrics, they prioritize performance over an understanding of the underlying signals.

In contrast, we treat pre-merge signals as a *discovery mechanism*. By optimizing linear combinations of 28 pairwise metrics, we prioritize transparency over non-linear complexity to identify properties driving success across merging methods. We even found that more complex non-linear methods, such as MLPs, underperform simple linear combinations in our particular setting.

## 3. Mergeability Metrics

To study what drives mergeability before performing the merge, we introduce a set of 28 interpretable pairwise metrics that capture the geometric and functional relationship between two models and provide a pre-merge signal of mergeability. Because no single metric is sufficiently predictive in isolation (A.2), we combine them to characterize mergeability. These metrics are computed offline without performing the merge, enabling efficient pre-screening of model pairs. For layer-wise metrics, we report the mean across all layers. We categorize these metrics into the following five groups (detailed definitions in A.4).

### 3.1. Task Vector Geometry Metrics

Given two task vectors $\boldsymbol{\tau}_A, \boldsymbol{\tau}_B \in \mathbb{R}^D$, where $D$ is the number of parameters and $\boldsymbol{\tau}_i = \text{flatten}(\boldsymbol{\theta}_i - \boldsymbol{\theta}_0)$ for $i \in \{A, B\}$, with $\boldsymbol{\theta}_i$ denoting the fine-tuned weights for task $i$ and $\boldsymbol{\theta}_0$ the

pretrained weights, we examine their geometric relationship in weight space. The intuition is that task vectors pointing in similar directions may interfere less during merging.

- **Dot Product & Cosine Similarity**: Measure the directional alignment between task vectors, with and without magnitude consideration.

- $L_2$ **Distance**: Smaller values indicate similar updates.

- **Angle**: The directional difference in degrees.

- **Magnitude Ratio**: The ratio between task vector magnitudes; a value near 1 suggests balanced contributions.

### 3.2. Effective Rank Metrics

Recent work suggests that Task Arithmetic benefits from low-rank task vectors (Liu et al., 2025). We analyze this property using the singular values $\sigma_1 \geq \sigma_2$ of the $2 \times D$ matrix $\mathbf{T} = [\boldsymbol{\tau}_A; \boldsymbol{\tau}_B]^\top$ formed by the two task vectors.

- **Effective Rank (Global/Layer-wise)**: Measures singular value entropy; a value of 1 indicates low-rank alignment, while higher values suggest diffuse, conflicting subspaces. Layer-wise results are averaged across layers, weighted by the update magnitudes.

- **Effective Rank Score (Global/Layer-wise)**: A normalized version of the effective rank mapping values to $[1, 0]$, where higher scores benefit mergeability.

- **Stable Rank**: An alternative dimensionality measure based on the singular values' squared Frobenius norm.

- **Spectral Gap**: The normalized difference between singular values; a large gap indicates a single dominant direction, suggesting strong alignment.

- **Singular Value Ratio**: The ratio $\sigma_2/\sigma_1$, where smaller values signify stronger alignment.

### 3.3. Subspace Overlap Metrics

These metrics analyze how the principal directions modified by each task relate to one another, utilizing the Singular Value Decomposition (SVD) of individual $2D$ weight matrices to extract left ($U$) and right ($V$) singular vectors. Non-$2D$ parameters, such as biases, are ignored.

- **Singular Value Overlap**: The cosine similarity between the top-100 normalized singular value distributions, providing a measure of spectral similarity averaged across layers.

- **Left Subspace Overlap**: Measures the alignment between the top-k left singular vectors using the Frobenius norm of their product; we set $k = 10$ here and in subsequent metrics.

- **Right Subspace Overlap (Top-k and Bottom-k)**: Measures overlap between the strongest and weakest

right singular vectors, as different merging methods may be sensitive to different spectral components.

- **Interaction Matrix Overlap (Top-k and Bottom-k)**: Analyzes the principal angles between subspaces by reporting the mean squared singular value of the interaction matrix formed by the right singular vectors.

## 3.4. Activation-Based Metrics

To complement weight-space analysis with functional insights, we measure how similarly models process identical inputs. We use a calibration set $\mathcal{D}_{\text{cal}}$, unified from 10 random samples per task, to extract activations from the final encoder layer.

- **Activation $L_2$ Distance**: The Euclidean distance between mean activations across calibration samples.
- **Activation Cosine Similarity**: The directional alignment of mean activation vectors.
- **Activation Magnitude Ratio**: The ratio of activation norms, indicating consistency in feature scales.
- **Activation Dot Product**: A combined measure of direction and magnitude for activation alignment.

## 3.5. Gradient-Based Metrics

Gradient alignment often correlates with multitask learning success (Yu et al., 2020). With task-specific calibration data (10 random samples per task), we measure gradient similarity at two levels:

**Encoder Gradient Metrics.** Measure the similarity of loss gradients with respect to encoder parameters:

- **Encoder Gradient Cosine Similarity**: Measures the alignment of directions in which models require parameter updates.
- **Encoder Gradient $L_2$ Distance**: The Euclidean magnitude of the difference between gradient vectors.
- **Encoder Gradient Dot Product**: Captures joint directional and magnitude agreement.

**Input Gradient Metrics.** Capture how models attend to input features by computing gradients with respect to them:

- **Input Gradient Cosine Similarity**: Determines whether models focus on similar input regions.
- **Input Gradient $L_2$ Distance**: Quantifies the magnitude of input sensitivity differences.
- **Input Gradient Dot Product**: A combined measure of direction and magnitude for saliency alignment.

Table 1 summarizes all 28 metrics. Weight-space and effective-rank metrics depend only on task vectors and are

therefore the most computationally efficient. Activation- and gradient-based metrics additionally require a small calibration set and forward/backward passes, but they capture functional similarities that purely weight-space metrics may overlook. An in-depth cost analysis is provided in Appendix A.5. Importantly, the metric set is modular and can be expanded or pruned without changing the framework.

## 4. Methodology

To assess the predictive power of multi-metric combinations, we optimize a linear model to maximize the Pearson correlation between a predicted score and the actual post-merge normalized accuracy. Specifically, for $K = 28$ normalized metrics $\tilde{m}_{AB}^{(k)}$, we seek coefficients $\mathbf{w} = (w_1, \ldots, w_K)^\top$ such that the score

$$\sum_{k=1}^{K} w_k \cdot \tilde{m}_{AB}^{(k)} \tag{1}$$

best correlates with the post-merge normalized accuracy $p_{AB}$, where $\mathcal{T}_A$ and $\mathcal{T}_B$ refer to tasks $A$ and $B$. Figure 1 depicts the overall optimization procedure.

Each training and validation sample is a tuple $\langle \mathcal{T}_A, \mathcal{T}_B, F, p_{AB} \rangle$, representing task $A$, task $B$, the merging method $F$, and the resulting post-merge normalized accuracy $p_{AB}$. In Appendix 7.2, we ablate linear optimization against a non-linear alternative, the MLP, proving the linear approach to be superior in stability, interpretability, and even performance, especially given the limited and noisy nature of the dataset. Thus, we regard our linear optimization framework to be the optimal choice for this particular setting of mergeability prediction.

**Metric Normalization.** To ensure fair contribution to the linear combination, we apply min-max normalization to map all metrics to the range $[-1, 1]$ to account for varying original scales (e.g., bounded cosine similarity vs. unbounded $L_2$ distances). Minima and maxima are computed across the training data to avoid data leakage during validation.

**Optimization Objective.** We formulate the coefficient learning problem as maximizing the Pearson correlation coefficient between predicted mergeability scores and observed post-merge normalized accuracy $p$, regularized by an L1 penalty to encourage sparsity:

$$\mathbf{w}^* = \arg\max_{\mathbf{w}} \ r(\mathbf{M}\mathbf{w}, \mathbf{p}) - \lambda \|\mathbf{w}\|_1 \tag{2}$$

where $\mathbf{p} \in \mathbb{R}^N$ is the vector of average normalized classification accuracies of the merged model across all $N$ pairs of tasks, relative to individual models (i.e., $\frac{1}{2} \left[ \frac{\text{acc}_{\text{merge}, A}}{\text{acc}_A} + \frac{\text{acc}_{\text{merge}, B}}{\text{acc}_B} \right]$), $\mathbf{M} \in \mathbb{R}^{N \times K}$ is the matrix of normalized metrics for $N$ model pairs, $r(\cdot, \cdot)$ denotes the Pearson correlation,

and $\lambda > 0$ controls the sparsity of the solution, we set $\lambda = 1.0$ by default without further tuning.

The L1 penalty serves two purposes. First, it performs automatic feature selection: with 28 candidate metrics, some of which are correlated, an unconstrained solution would distribute weight diffusely and obscure which metrics genuinely drive mergeability. L1 regularization identifies a compact, interpretable subset of predictors per merging method. Second, it acts as mild regularization to prevent overfitting, given the limited data we describe later.

We optimize for Pearson correlation rather than MSE because our goal is to *rank* task pairs by mergeability, not to predict exact accuracies. Correlation is scale-invariant, which is desirable since the linear combination of features lives on an arbitrary scale, while accuracies lie in a narrow range. Optimizing MSE, which we will ablate, would entangle learning the correct ordering with matching this scale and perform poorly. In contrast, high correlation directly reflects the ranking quality practitioners care about: given candidate task pairs, which should be merged first? It also yields a single interpretable metric, the correlation coefficient $r$, summarizing how much variance in merge outcomes is explained by the learned combination.

**Optimization Procedure.** The coefficients are optimized via Adam (Kingma & Ba, 2014) with a learning rate of 0.01, minimizing the negative Pearson correlation plus the L1 penalty ($\lambda = 1.0$). We use early stopping with a patience of 50 iterations, halting when the improvement in training correlation falls below $10^{-4}$.

**Leave-One-Task-Out Cross-Validation (LOTO).** A key challenge in evaluating mergeability prediction is ensuring that the learned coefficients generalize to unseen task combinations. Simple random train-validation splits may still contain pairs involving tasks seen during training, leading to optimistic estimates. To address this, we employ *leave-one-task-out* (LOTO) cross-validation.

Let $\mathcal{T} = \{T_1, \ldots, T_M\}$ denote the set of $M$ tasks. In each fold $f$, we hold out all pairs involving task $T_f$:

$$\mathcal{V}_f = \{(i, j) : T_i = T_f \text{ or } T_j = T_f\} \quad (3)$$

The remaining pairs form the training set $\mathcal{S}_f$. We optimize coefficients $\mathbf{w}_f$ on $\mathcal{S}_f$ and evaluate on $\mathcal{V}_f$. This ensures that the validation pairs involve at least one task never seen during training, providing a rigorous test of generalization to novel task combinations.

For $M$ tasks with $\binom{M}{2}$ total pairs, each fold holds out the $M - 1$ pairs involving the held-out task. We report: (i) *per-fold metrics*, including training and validation correlations for each fold; and (ii) *average coefficients*, $\bar{\mathbf{w}} = \frac{1}{M} \sum_{f=1}^{M} \mathbf{w}_f$ with standard deviations, to identify metrics that consistently contribute to prediction across folds.

This cross-validation scheme is more stringent than random splitting: rather than testing on random held-out pairs (which may involve familiar tasks in new combinations), we test on pairs where at least one task is entirely novel. High validation correlation under LOTO provides strong evidence that the learned metric combination captures genuine mergeability signals rather than task-specific artifacts.

## 5. Experimental Setup

**Tasks and Models.** Following Gargiulo et al. (2025), we evaluate our mergeability discovery framework on 20 image classification tasks. See the full list of tasks in Appendix A.16. We fine-tune CLIP ViT-B/16 and B/32 (Radford et al., 2021). see Appendix A.15.5) for the fine-tuning setup. This yields 20 task-specific models, from which we extract task vectors $\boldsymbol{\tau}_i = \boldsymbol{\theta}_i - \boldsymbol{\theta}_0$ for each task $i$. This ultimately yields $\binom{20}{2} = 190$ unique model pairs. We choose ViT-B/16 as the default model for analysis when B/32 is not explicitly mentioned.

**Merge Methods.** We evaluate five representative merging methods that span different algorithmic approaches:

1. **Task Arithmetic (TA)** (Ilharco et al., 2023): Adds scaled task vectors to the pretrained model: $\boldsymbol{\theta}_{\text{merged}} = \boldsymbol{\theta}_0 + \lambda(\boldsymbol{\tau}_A + \boldsymbol{\tau}_B)$.
2. **Weight Averaging (WA)** (Wortsman et al., 2022): Simple average of fine-tuned weights.
3. **Task Singular Vectors (TSV)** (Gargiulo et al., 2025): Orthogonalizes overlapping task singular directions in task vectors to minimize interference.
4. **TIES** (Yadav et al., 2023): Resolves sign conflicts and trims low-magnitude parameters to reduce task vector interference before merging.
5. **DARE** (Yu et al., 2023): Randomly drops and rescales task vector parameters prior to merging, reducing parameter interference.

## 6. Results

Under the LOTO protocol, upon optimizing the linear coefficients for all 20 folds with gradient descent, we obtain the cross-validation results for all methods, as reported in Table 2. We report the per-fold means of the Pearson correlation between the predicted post-merge normalized accuracy and the actual value, across all folds. We observe different degrees of predictivity across model-merger combinations, with ViT-B/32 exhibiting more predictable mergeability than its larger B/16 counterpart. The actual learned coefficients of the linear optimization are visually provided in Figure 2 and examined in depth in Appendix A.3. Next, we discuss observations that emerged from these results.

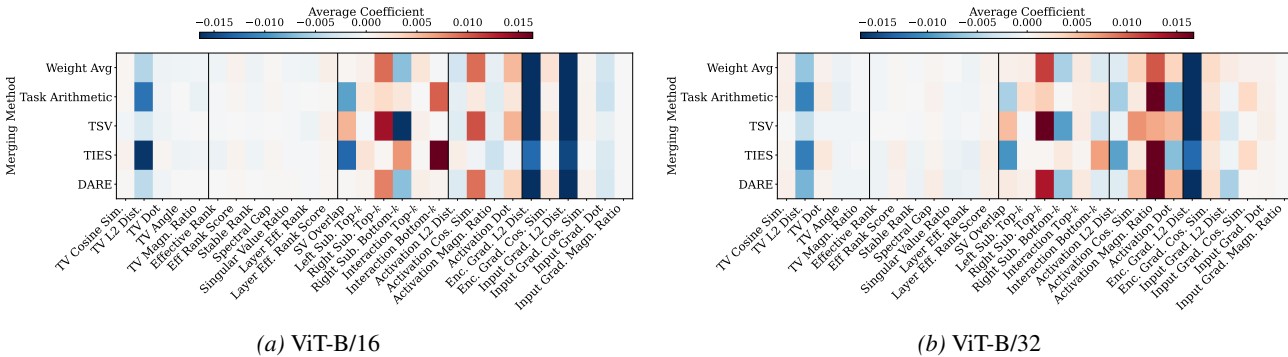

*Figure 2.* Average learned $L_1$-regularized LOTO coefficients across folds for ViT-B/16 and B/32. Vertical lines delimit metric categories.

*Table 2.* Leave-one-task-out cross-validation results across the two backbones. Values show mean Pearson correlation $r$ across 20 folds for the training and validation sets, using L1-regularized linear predictors ($\lambda = 1.0$). The *Average* row reports the mean across the five mergers.

| Method | Per-Fold Pearson Correlation ($r$) | |
|---|---|---|
| | Train | Val |
| *ViT-B/16* | | |
| Task Arithmetic | 0.574 | 0.445 |
| Weight Averaging | 0.702 | 0.589 |
| TSV | 0.754 | 0.627 |
| TIES | 0.552 | 0.463 |
| DARE | 0.722 | 0.596 |
| **Average** | *0.661* | *0.544* |
| *ViT-B/32* | | |
| Task Arithmetic | 0.739 | 0.619 |
| Weight Averaging | 0.799 | 0.699 |
| TSV | 0.784 | 0.692 |
| TIES | 0.671 | 0.543 |
| DARE | 0.801 | 0.684 |
| **Average** | *0.759* | *0.647* |

**Mergeability predictability varies across methods.** TSV merging exhibits the highest overall predictability, with a per-fold validation correlation of 0.627 and 0.692 across the two backbones. Weight averaging follows with 0.589 and 0.699, and DARE is just slightly behind. Task Arithmetic and TIES exhibit perceivably lower validation correlations.

**Generalization gap.** All methods exhibit a gap between training and validation correlations, which is expected given that LOTO tests on unseen task combinations. The gap is, on average, a relative drop of 17.7% on ViT-B/16 and 14.7% on ViT-B/32. Across both backbones, *TSV and WA are the most stable generalizers*: TSV exhibits the smallest gap on both ($0.754 \rightarrow 0.627$ on B/16; $0.784 \rightarrow 0.692$ on B/32).

**Method-specific patterns.** Method-specific patterns reveal two distinct clusters of success fingerprints. *Weight Averaging, TSV, and DARE* rely on a nearly identical top-5 (100%)

overlap): gradient $L_2$ distances (`encoder gradient L2`, `input gradient L2`, both negative), subspace overlap (`right subspace overlap top-k`, positive), and activation alignment (`activation cosine similarity` or `activation dot product`, positive). In contrast, *Task Arithmetic and TIES* form a separate cluster that also shares 100% top-5 overlap but draws predominantly on task-vector geometry and subspace *bottom-k* structure: `task vector L2 distance` (negative), `interaction matrix overlap bottom-k` (positive), and `singular value overlap` (negative), alongside the gradient $L_2$ distances. Notably, for TIES, `interaction matrix overlap bottom-k` is among the largest positive coefficients ($+0.015$), indicating that its sign-conflict resolution particularly benefits from agreement on low-rank residual directions.

**Gradient and subspace metrics are consistent signals.** Across all five merging methods and both backbones, the $L_2$ distances of encoder and input gradients receive the largest-magnitude negative coefficients (e.g., $-0.021$ and $-0.018$ on ViT-B/16 averaged across methods), confirming that *gradient discrepancies consistently correlate to poorer merge outcomes* ([Daheim et al., 2024](#)). Simultaneously, at least one subspace-overlap metric appears in every method's top-10, with `right subspace overlap top-k` receiving a consistently positive coefficient ($+0.007$ on average). Together, these form a stable, method-agnostic core: *low gradient discrepancy and high principal-subspace alignment are prerequisites for mergeability regardless of the merging algorithm*. Successful merging thus depends strongly on optimization dynamics (gradient similarity) and moderately on representational geometry (subspace structure).

**Statistical Stability** For the dominant metrics, the per-fold standard deviation is comparable to or smaller than than the coefficient mean (e.g., `encoder gradient L2`: $-0.025 \pm 0.011$ for WA; $-0.026 \pm 0.008$ for TSV), indicating that the identified fingerprints are not artifacts of any particular held-out task. We also repeated the experiments for ViT-B/16 with 10 different random seeds to

confirm reproducibility and stability (see Appendix A.8).

**Takeaway.** These findings demonstrate that *mergeability is method-dependent*: pairs of models that merge well under one algorithm may fail under another, and each method weights mergeability signals differently. Yet the five methods split into just two fingerprint clusters, {WA, TSV, DARE} and {TA, TIES}, and both clusters share a common, method-agnostic core of gradient and subspace alignment. Practitioners can therefore exploit two levels of diagnostic information: the universal core to screen candidate pairs across any merger, and the method-specific fingerprint to choose which merger best suits a given pair. We investigate this stable, method-agnostic core in the following section.

### 6.1. Stable Metrics

While we just showed that optimal metric weightings are method-dependent, we now examine whether certain metrics exhibit *stable* predictive behavior across all merging methods *and* architectures. We define a stable metric as one whose average LOTO coefficient maintains a consistent sign across all five methods on both ViT-B/16 and ViT-B/32 (10 model–method combinations), indicating mergeability principles that transcend specific merging algorithms and backbone sizes. To identify such metrics, we analyze the average coefficients $\bar{w}_k$ over the LOTO folds for each combination; stable metrics are reported in Table 3.

*Table 3.* Stable predictive metrics and their average coefficients ($\times 10^{-3}$), averaged across ViT-B/16 and ViT-B/32.

| Metric | WA | TA | TSV | TIES | DARE | AVG |
|---|---|---|---|---|---|---|
| Encoder Grad. $L_2$ Distance | −28.0 | −19.8 | −26.8 | −12.9 | −30.2 | −23.5 |
| Task Vector $L_2$ Distance | −5.6 | −11.7 | −3.2 | −13.8 | −6.0 | −8.1 |
| Right Subspace Overlap Top-k | +10.4 | +3.5 | +15.7 | +0.3 | +10.9 | +8.2 |
| Encoder Grad. Cosine Sim. | +2.7 | +1.4 | +2.4 | +1.0 | +2.4 | +2.0 |
| Input Grad. Cosine Sim. | +0.6 | +2.0 | +0.3 | +1.7 | +0.5 | +1.0 |

**Gradient Distance as a Consistent Indicator.** The most dominant stable metric is `encoder gradient` $L_2$ `distance`, which carries a strong negative coefficient across all methods and both architectures (averaging $-23.5 \times 10^{-3}$). This indicates that *models with similar encoder gradients merge better*, regardless of the merging algorithm or backbone size. The encoder gradient captures how each model's loss responds to changes in the encoder's representations; similarity in this response suggests that the two models have learned compatible solutions. Complementing this, `encoder gradient cosine similarity` is stably positive (avg $+2.0 \times 10^{-3}$) and `input gradient cosine similarity` is stably positive (avg $+1.0 \times 10^{-3}$). Together, these three gradient-based stable metrics tell a consistent story: models whose gradient landscapes are both close in magnitude *and* aligned in direction merge more reliably, a finding consistent with Daheim et al. (2024),

where merging failure is associated with gradient mismatch.

**Task Vector Similarity and Subspace Alignment.** `Task vector` $L_2$ `distance` carries a stable negative coefficient (avg $-8.1 \times 10^{-3}$), indicating that pairs with similar task vectors, smaller displacement from the pretrained model in similar directions, merge more readily. This effect is most pronounced for TIES and Task Arithmetic (both $\approx -12$ to $-14$), which directly operate on the task vector magnitudes. `Right subspace overlap top-k` is stably positive (avg $+8.2 \times 10^{-3}$): alignment in the top-k right singular vectors of the task vectors, which encode the most impacted input-space directions, promotes mergeability. The effect is strongest for TSV ($+15.7$) and weaker for TIES ($+0.3$), where task vector geometry dominates.

**Interpreting Stable Metrics Through the Lens of Task Interference.** The five stable metrics can be unified under the concept of *task interference*. Model merging fails when the combined task vectors interfere destructively (Yadav et al., 2023; Daheim et al., 2024; Wang et al., 2024; Deep et al., 2024; Zhou et al., 2025a), either by pointing in opposing directions or by competing for the same representational capacity (Yang et al., 2024). Our stable metrics capture complementary aspects of this interference:

- **Gradient metrics** measure functional interference: do the tasks require conflicting parameter updates? Low $L_2$ distance and high cosine similarity between encoder and input gradients both indicate that the two fine-tuned models respond to data in compatible ways.

- **Subspace metrics** measure structural interference: do the tasks modify overlapping or orthogonal directions in weight space? High `right subspace overlap top-k` indicates that the dominant input-space directions modified by each task vector are aligned, reducing destructive interference.

- **Task vector geometry** measures skill overlap: close task vectors (low $L_2$ distance) imply that the two tasks have reached similar task-solving solutions from the pretrained base, a favorable condition for merging.

**Practical Implications.** The existence of stable metrics has important practical implications:

i) **Method-Agnostic Screening**: Before selecting a merging algorithm, practitioners can compute the stable metrics to assess whether two models satisfy fundamental compatibility constraints. Low `encoder gradient` $L_2$ `distance`, low `task vector` $L_2$ `distance`, and high `right subspace overlap top-k` indicate that the pair avoids severe task interference and is therefore *eligible* for successful merging under most methods. Because mergeability is also method-dependent, these stable

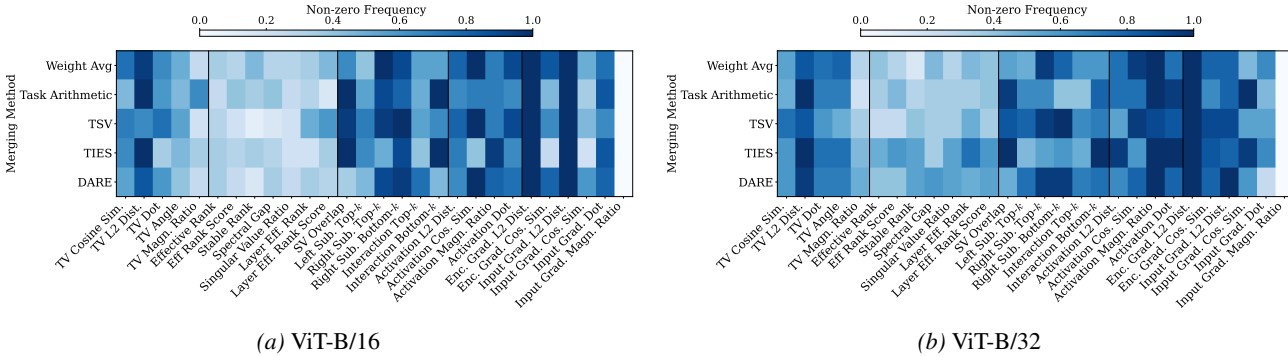

*(a)* ViT-B/16                                    *(b)* ViT-B/32

*Figure 3.* Fraction of folds in which each metric's $L_1$-regularized LOTO coefficient is non-zero. Vertical lines delimit metric categories.

metrics should be interpreted as necessary but not sufficient conditions: satisfying them does not guarantee strong merge performance, but not satisfying them leads to severe degradation.

ii) **Training Guidance**: The importance of gradient-based metrics suggests that fine-tuning procedures that promote gradient similarity across tasks (e.g., via explicit regularization or shared optimization trajectories) benefit model mergeability. When the merging method is known a priori, one could enforce method-specific constraints during fine-tuning to improve mergeability. We discuss a proof-of-concept experiment next.

**Enhancing Mergeability during fine-tuning.** To demonstrate the actionability of our findings, we conduct a *toy experiment* motivated by the importance of gradient distance. We fine-tune task-specific ViT-B/16 models with a gradient-magnitude penalty, encouraging smaller parameter steps and keeping models closer to the pretrained solution. Operating within a shared loss basin naturally promotes gradient alignment. Consistent with this intuition (Ortiz-Jimenez et al., 2024; Zhou et al., 2025a), the regularized models are more mergeable, yielding a modest but meaningful improvement in post-merge normalized accuracy (average $\Delta = +0.41$). See Appendix A.12 for details. We emphasize that this experiment serves as a proof of concept; more principled fine-tuning strategies, targeting multiple signals and specific merging methods, are expected to yield larger gains.

## 7. Ablations

To validate our framework's architectural design, we perform a series of ablation studies evaluating our primary design choices and the contribution of different metric categories on mergeability outcomes.

### 7.1. Effect of $L_1$ Regularization

We adopt $L_1$-regularized optimization ($\lambda = 1.0$) as our default, as it produces sparser, more stable predictors. To

quantify the benefit, we compare against an unregularized baseline ($\lambda = 0$). As depicted in Table 19 (Appendix A.13), $L_1$ regularization reduces prediction variance substantially: the average fold-to-fold standard deviation of validation correlation drops from 0.254 to 0.159 on ViT-B/16 and from 0.201 to 0.164 on ViT-B/32. In terms of predictive accuracy, $L_1$ matches or outperforms the dense baseline on ViT-B/16 (0.544 vs. 0.525), while on ViT-B/32 the dense model achieves marginally higher average validation correlation (0.676 vs. 0.648) but at the cost of higher variance and nearly double the number of active features (27.8 vs. 18.5). Overall, $L_1$ regularization achieves better or comparable accuracy with roughly 60% of the features (17–19 vs. 27–28), while consistently eliminating effective-rank metrics and retaining subspace and gradient-based metrics across folds. For an intuitive visual understanding, Figure 3 depicts the non-zero frequencies of all metrics in our L1 linear optimization, across the 20 folds (discussion in Appendix A.13). The overall predictive advantage and stability of the L1 approach, across both backbones, confirms that the identified stable predictors reflect a reliable, robust signal of mergeability rather than an artifact of coefficient scaling across the full redundant feature set.

### 7.2. L1 Linear Optimization vs MLP

To assess whether non-linear predictors can capture relationships missed by the linear model, we compare our L1-regularized linear optimization against a per-method MLP (single hidden layer, 8 units, dropout $p = 0.4$, 300 epochs, same LOTO protocol). The linear model wins consistently: on ViT-B/16, L1 achieves an average validation $r = 0.544$ vs. 0.493 for the MLP. The L1 linear model outperforms the MLP on 4/5 methods, while performing on par on one. We attribute this to the limited training set size (∼170 training pairs per fold), the approximately linear structure of the mergeability signal, and the implicit feature selection provided by the $L_1$ penalty. A more comprehensive discussion of the ablation can be found in Appendix A.7.

*Table 4.* Full metric category ablation results. Validation Pearson correlation ($r$) reported as mean across LOTO folds;

| | Method | Full Set (28) | No Subspace (−6) | No Gradient (−6) | No EffRank (−7) | No TaskVec (−5) | No Activ (−4) |
|---|---|---|---|---|---|---|---|
| ViT-B/16 | Weight Avg | 0.589 | 0.599 (+0.010) | 0.382 (−0.207) | 0.622 (+0.032) | 0.607 (+0.017) | 0.596 (+0.006) |
| | Task Arithmetic | 0.445 | 0.404 (−0.041) | 0.253 (−0.192) | 0.462 (+0.017) | 0.442 (−0.002) | 0.462 (+0.017) |
| | TSV | 0.627 | 0.612 (−0.015) | 0.532 (−0.095) | 0.630 (+0.003) | 0.662 (+0.035) | 0.612 (−0.015) |
| | TIES | 0.463 | 0.293 (−0.170) | 0.365 (−0.098) | 0.492 (+0.028) | 0.380 (−0.083) | 0.473 (+0.010) |
| | DARE | 0.596 | 0.585 (−0.011) | 0.441 (−0.155) | 0.617 (+0.022) | 0.591 (−0.005) | 0.586 (−0.010) |
| | **Avg** $\triangle$ | — | −0.045 | **−0.149** | +0.020 | −0.008 | +0.002 |
| ViT-B/32 | Weight Avg | 0.699 | 0.638 (−0.061) | 0.537 (−0.162) | 0.738 (+0.039) | 0.646 (−0.053) | 0.682 (−0.018) |
| | Task Arithmetic | 0.619 | 0.632 (+0.013) | 0.522 (−0.097) | 0.613 (−0.006) | 0.608 (−0.011) | 0.483 (−0.136) |
| | TSV | 0.692 | 0.637 (−0.055) | 0.511 (−0.181) | 0.688 (−0.004) | 0.684 (−0.008) | 0.688 (−0.004) |
| | TIES | 0.543 | 0.539 (−0.004) | 0.492 (−0.050) | 0.554 (+0.012) | 0.547 (+0.004) | 0.436 (−0.107) |
| | DARE | 0.684 | 0.723 (+0.038) | 0.519 (−0.165) | 0.707 (+0.022) | 0.689 (+0.004) | 0.654 (−0.031) |
| | **Avg** $\triangle$ | — | −0.014 | **−0.131** | +0.012 | −0.013 | −0.059 |

### 7.3. Pearson Correlation vs MSE as Objective

We also compare our Pearson-correlation objective against mean squared error (MSE) minimization. Since our goal is to *rank* task pairs by mergeability rather than predict exact performance values, correlation-based optimization is a natural fit: it is invariant to the scale and shift of predictions and focuses purely on relative ordering. The results confirm this decisively. On ViT-B/16, Pearson+L1 achieves average validation $r = 0.544$ versus $0.147$ for MSE ($3.7\times$ improvement); on ViT-B/32, $0.648$ vs. $0.143$ ($4.5\times$). MSE degrades because it must simultaneously learn the correct ranking and the correct output scale, wasting capacity on the latter. For full results, refer to Appendix A.6.

### 7.4. Metric Category Ablation

To validate the coefficient-based analysis, we perform ablation experiments by removing entire metric categories from the L1-regularized linear optimization and measuring the resulting change in validation Pearson correlation. Main conclusions are consistent across ViT-B/16 and ViT-B/32, and Appendix A.14 illustrates the full details.

As reported in Table 4, gradient-based metrics are the most critical category: removing them always causes the largest performance drop on both backbones (average $\Delta r = -0.149$ on ViT-B/16 and $-0.131$ on ViT-B/32), with all five merging methods affected. This is consistent with the stable-metric analysis, where `encoder gradient` $L_2$ `distance` exhibited the strongest and most consistent coefficient. Subspace metrics are the second most important category, though with greater method and architecture dependence: on ViT-B/16, removing subspace metrics reduces performance most sharply for TIES ($\Delta r = -0.170$), while on ViT-B/32 the effect is distributed across Weight Averaging ($-0.061$) and TSV ($-0.055$). Activation-based metrics are comparably important on average, but their contribution is architecture-dependent: near-zero for ViT-B/16 ($\Delta r = +0.002$), yet substantial for ViT-B/32 ($\Delta r = -0.059$), particularly for Task Arithmetic ($-0.136$) and

TIES ($-0.107$). Task vector geometry metrics contribute marginally ($\Delta r \approx -0.01$) to both backbones. Effective-rank metrics are the only category whose removal consistently *improves* validation performance ($+0.020$ on ViT-B/16, $+0.012$ on ViT-B/32), confirming that they introduce noise rather than signal and are effectively suppressed by the $L_1$ regularization.

This outcome demonstrates the framework's ability to isolate dominant signals and identify redundancy without prior knowledge, regardless of the merging method and tasks.

## 8. Conclusion

We presented an interpretable framework for mergeability that treats merge success as a function of pairwise geometric and functional properties, rather than as an intrinsic attribute of individual models. Using an L1-regularized linear predictor over 28 pairwise metrics and evaluating it with leave-one-task-out (LOTO) cross-validation, the framework achieves Pearson correlations of $r \in [0.45, 0.70]$ across five merging methods and two ViT backbones, showing generalization to unseen tasks. Our analysis further suggests that mergeability is inherently method-dependent: each algorithm exhibits a distinct success fingerprint. At the same time, a small set of stable metrics persists across all methods and both architectures, with gradient-based signals emerging as the most consistent method-agnostic indicators of mergeability. Metric-category ablations reinforce this picture: gradient-based metrics contribute the most predictive power overall, whereas effective-rank metrics are largely redundant. On the practical side, the stable metrics provide a lightweight pre-merge screening tool for identifying broadly compatible model pairs before merging, while the method-specific fingerprints inform properties to target during fine-tuning to improve mergeability under the chosen method. Our proof-of-concept regularization experiment shows that the discovered signals are also actionable: encouraging gradient similarity during fine-tuning improves post-merge performance for most methods.

## Acknowledgements

This work is supported by the MUR FIS2 grant n. FIS-2023-00942 "NEXUS" (cup B53C25001030001), and partly by Sapienza University of Rome via the Seed of ERC grant "MINT.AI" (cup B83C25001040001).

RY and BZ were supported in part by the U.S. Army Research Office under Army-ECASE award W911NF-07-R-0003-03, the U.S. Department Of Energy, Office of Science, IARPA HAYSTAC Program, and NSF Grants #2205093, #2146343, #2134274, #2441832, CDC-RFA-FT-23-0069, DARPA AIE FoundSci and DARPA YFA.

## Impact Statement

This paper presents work whose goal is to advance the field of Machine Learning by providing an interpretable framework for predicting the success of model merging. By identifying specific pairwise properties, such as gradient distance and subspace overlap, that determine mergeability, our research enables practitioners to combine task-specific models more reliably without the need for expensive joint retraining.

From a societal and ethical perspective, this work contributes to more sustainable AI development by reducing the computational resources and energy consumption required to create multi-task systems. Furthermore, by moving away from black-box predictors toward a transparent, linear model of mergeability, we enhance the interpretability of model interactions. This transparency is a critical step toward understanding and mitigating potential failures or performance degradations in merged models used in downstream applications.

## LLM Usage Statement

The authors leveraged LLMs to assist in the following tasks:

- **Polishing the paper:** Refining the language for clarity, correcting grammatical errors, and ensuring stylistic consistency throughout the manuscript.
- **Assisting in code writing:** Aiding in the development and debugging of the scripts used for the experimental evaluation of model mergeability.
- **Writing code to generate LaTeX tables:** Automating the conversion of raw experimental data into LaTeX table syntax and refining complex formatting, such as multi-row alignment and column spacing.

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

# A. Appendix

## A.1. Limitations

While our framework demonstrates robust generalization across five merging methods and two ViT architectures, we identify several avenues for further refinement. First, although our 20-task benchmark provides 190 pairwise evaluations, the most extensive of its kind, curating such a diverse suite of tasks is resource-intensive and may pose challenges in hardware-constrained environments. Second, our reliance on activation- and gradient-based metrics necessitates a small set of calibration data, which precludes application in strictly data-free scenarios. Third, our current analysis focuses on pairwise merging; extending this framework to $k$-way mergeability ($k > 2$) remains a significant challenge due to the complex, non-linear interactions between multiple model checkpoints. Fourth, while this study is cross-architectural, it focuses on vision classifiers; models in other domains (e.g., natural language processing or audio) may exhibit different patterns in metric importance. However, our framework is designed to be domain-agnostic and remains applicable to any scenario where pretrained and fine-tuned checkpoints are available, and activations and gradients can be extracted.

Regarding our predictive modeling, we found that a linear predictor outperformed an MLP. We attribute this to the relatively modest scale of the training data and a predominantly linear signal structure in the current feature space; however, this does not preclude the utility of non-linear models as dataset scales increase. Finally, while our regularization experiments demonstrate improved mergeability across four out of five methods, the lack of universal gain suggests that more sophisticated, method-aware fine-tuning strategies may be required to maximize performance across all model merging regimes.

## A.2. Individual Metric Correlations

To motivate the need for a learned linear combination of mergeability metrics, we examine the predictive power of each metric in isolation. Table 5 reports the Pearson correlation between each individual metric and the normalized post-merge accuracy (i.e., post-merge performance divided by task-specific model performance, averaged across the two tasks) across all 190 task pairs, computed separately for each merging method.

The results reveal that no single metric consistently achieves strong correlations across all merging methods. Activation-based metrics, particularly activation dot product ($r = 0.572$ for TSV, $r = 0.450$ for Weight Averaging, $r = 0.447$ for DARE) and activation cosine similarity ($r = 0.521$ for TSV, $r = 0.372$ for DARE, $r = 0.366$ for Weight Averaging), achieve the highest individual correlations for three of the five methods. However, these same metrics are weak for Task Arithmetic ($r = 0.094$, $r = 0.146$) and near-zero or slightly negative for TIES ($r = -0.083$, $r = 0.030$), indicating that their predictive utility is strongly method-dependent.

Task vector geometry metrics, which are computationally inexpensive and have been proposed as indicators of mergeability in prior work, show uniformly weak correlations (typically $|r| < 0.15$) across all methods, with the only significant exception being TSV task vector $L_2$ distance ($r = -0.211$). Similarly, effective rank metrics achieve at most weak correlations ($|r| < 0.2$), restricted to TSV.

Subspace overlap metrics exhibit a strikingly different pattern across methods. Bottom-$k$ variants (Right Sub Bot-$k$, Interact Bot-$k$) are the strongest individual predictors for TIES ($r \approx 0.28$) and moderately predictive for Task Arithmetic ($r \approx 0.21$), but near-zero or negative for Weight Averaging, TSV, and DARE. Conversely, singular value overlap carries a small but significant negative correlation for four methods (TA, WA, TIES, DARE) but is uninformative for TSV.

Gradient-based metrics (encoder/input $L_2$ distance) are moderately predictive for Weight Averaging ($r \approx -0.26$ to $-0.32$) and DARE ($r \approx -0.25$ to $-0.32$), but weak for TSV and near-zero for TIES, confirming that, in isolation, these metrics are not universal indicators either.

The inconsistency of individual metric correlations across methods underscores a key finding: mergeability prediction requires combining multiple complementary signals. A metric that is strongly predictive for one method (e.g., activation dot product for DARE at $r = 0.447$) may be uninformative or even mildly negative for another (TIES, $r = -0.083$), and metrics dominant for TIES (bottom-$k$ subspace overlaps) are nearly useless for DARE and Weight Averaging. This observation motivates our approach of learning method-specific weighted combinations of metrics, which achieves substantially higher correlations (see Section 6) by leveraging the complementary information captured by different metric categories.

*Table 5.* Pearson correlation between individual mergeability metrics and normalized post-merge accuracy. Cell shading indicates significance levels (Darkest: $p < 0.001$, Medium: $p < 0.01$, Lightest: $p < 0.05$), with no shade signifying statistical insignificance.

| | Metric | Task Arith. | Weight Avg | TSV | TIES | DARE |
|---|---|---|---|---|---|---|
| **Task Vector** | Task Vector Cosine Sim | 0.015 | 0.030 | 0.082 | -0.001 | 0.039 |
| | Task Vector $L_2$ Dist | -0.079 | -0.079 | -0.211 | -0.069 | -0.111 |
| | Task Vector Dot Prod | -0.015 | 0.020 | -0.002 | -0.031 | 0.010 |
| | Weight Angle | -0.017 | -0.032 | -0.082 | -0.001 | -0.041 |
| | Task Vector Mag Ratio | -0.037 | 0.057 | 0.092 | -0.061 | 0.046 |
| **Effective Rank** | Eff Rank | -0.009 | 0.116 | 0.177 | -0.060 | 0.106 |
| | Eff Rank Score | 0.009 | -0.116 | -0.177 | 0.060 | -0.106 |
| | Stable Rank | -0.014 | 0.104 | 0.157 | -0.061 | 0.094 |
| | Spectral Gap | 0.032 | -0.065 | -0.098 | 0.060 | -0.053 |
| | SV Ratio | -0.032 | 0.065 | 0.098 | -0.060 | 0.053 |
| | Layer Eff Rank | 0.015 | 0.136 | 0.180 | -0.041 | 0.126 |
| | Layer Eff Rank Score | -0.015 | -0.136 | -0.180 | 0.041 | -0.126 |
| **Sub. Overlap** | SV Overlap | -0.167 | -0.164 | 0.010 | -0.161 | -0.145 |
| | Left Sub Top-$k$ | 0.093 | -0.007 | -0.048 | 0.100 | -0.004 |
| | Right Sub Top-$k$ | 0.062 | -0.051 | -0.068 | 0.064 | -0.047 |
| | Right Sub Bot-$k$ | 0.211 | 0.009 | -0.079 | 0.281 | -0.007 |
| | Interact Top-$k$ | 0.071 | 0.006 | -0.002 | 0.063 | 0.010 |
| | Interact Bot-$k$ | 0.209 | 0.006 | -0.073 | 0.275 | -0.007 |
| **Activation** | Act $L_2$ Dist | -0.167 | -0.313 | -0.446 | -0.085 | -0.321 |
| | Act Cosine Sim | 0.146 | 0.366 | 0.521 | 0.030 | 0.372 |
| | Act Mag Ratio | 0.050 | -0.001 | -0.005 | 0.024 | 0.001 |
| | Act Dot Prod | 0.094 | 0.450 | 0.572 | -0.083 | 0.447 |
| **Gradient** | Enc Grad Cos | 0.012 | 0.086 | 0.056 | -0.026 | 0.085 |
| | Enc Grad $L_2$ | -0.099 | -0.257 | -0.111 | -0.061 | -0.246 |
| | Enc Grad Dot | 0.061 | 0.082 | -0.009 | 0.045 | 0.066 |
| | Input Grad Cos | -0.035 | -0.020 | -0.032 | -0.029 | -0.022 |
| | Input Grad $L_2$ | -0.162 | -0.322 | -0.174 | -0.093 | -0.317 |
| | Input Grad Dot | -0.067 | -0.083 | 0.002 | -0.056 | -0.076 |

## A.3. Learned Coefficient Values

While the main paper discusses the learned coefficients for all methods, here we provide a visual representation through coefficient heatmaps (Figures 4 and 5) and the exact numerical values in Table 8. Coefficients are averages across all 20 LOTO folds, using the L1-regularized predictor with $\lambda = 1.0$. They operate on min-max normalized metrics (scaled to $[-1, 1]$), with normalization statistics computed only on training data within each fold to prevent data leakage. Due to L1 regularization, many coefficients are shrunk to exactly zero; only the most predictive metrics per method retain non-zero weights. We use **ViT-B/16** as the default model for the subsequent analysis.

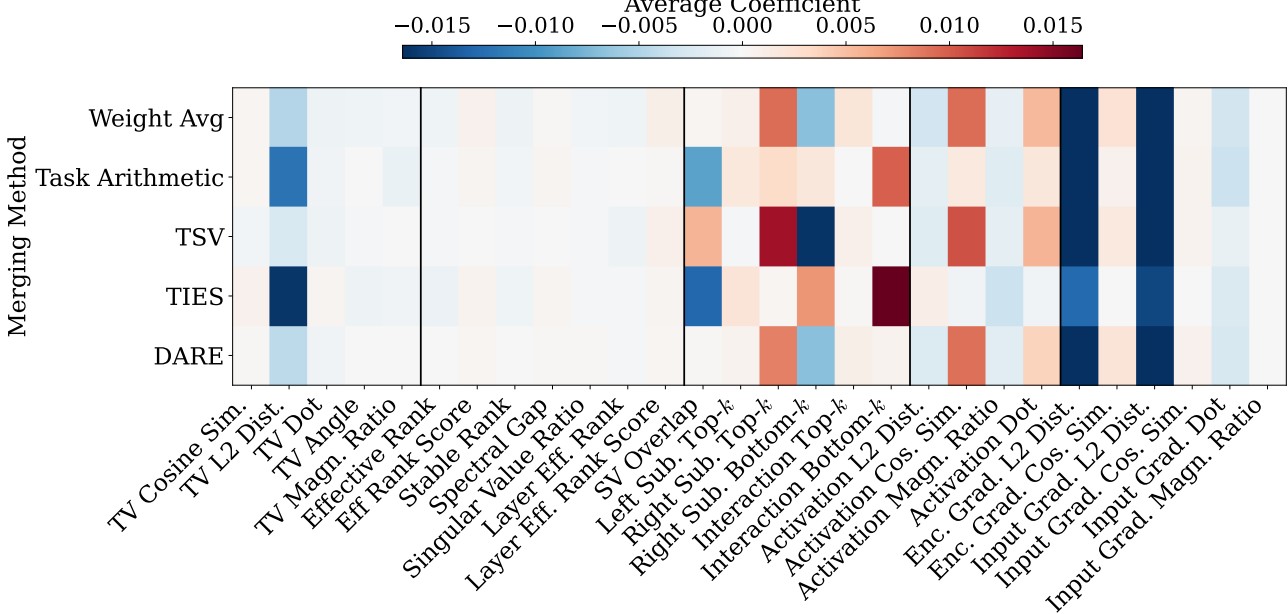

*Figure 4.* Per-method learned coefficients for all 28 metrics, averaged across 20 LOTO folds. **ViT-B/16** backbone.

**Interpretation.** A positive coefficient indicates that higher values of the corresponding metric predict better post-merge performance, while a negative coefficient indicates the opposite. The magnitude reflects predictive influence. Many coefficients are zero due to L1 sparsification; the non-zero subset constitutes the method's "success fingerprint".

**Dominant Metric Overlap.** We examine the overlap between the top-5 most dominant metrics (by average coefficient magnitude) across methods. Table 6 shows these overlaps for ViT-B/16. Average top-5 overlap is $0.64$, ranging from $0.4$ (e.g., TA–TSV) to $1.0$ (e.g., WA–TSV, WA–DARE, TSV–DARE). The clusters reflect algorithmic similarity: WA, TSV, and DARE share very similar fingerprints, while TA and TIES form a separate cluster with higher mutual overlap but lower overlap with the WA/TSV/DARE group.

*Table 6.* Overlap of top-5 metrics among methods (ViT-B/16). Average cross-method overlap is 0.64.

| Method | TA | WA | TSV | TIES | DARE |
|--------|-----|-----|-----|------|------|
| TA | 1.0 | 0.4 | 0.4 | 1.0 | 0.4 |
| WA | | 1.0 | 1.0 | 0.4 | 1.0 |
| TSV | | | 1.0 | 0.4 | 1.0 |
| TIES | | | | 1.0 | 0.4 |
| DARE | | | | | 1.0 |

**Sign Agreement.** Table 7 reports the percentage of sign agreement across the 28 metrics for each method pair. Average agreement is $0.79$, ranging from $0.57$ (TSV–TIES) to $0.93$ (WA–DARE). The TSV–TIES pair exhibits the lowest agreement, reflecting their fundamentally different merging mechanisms.

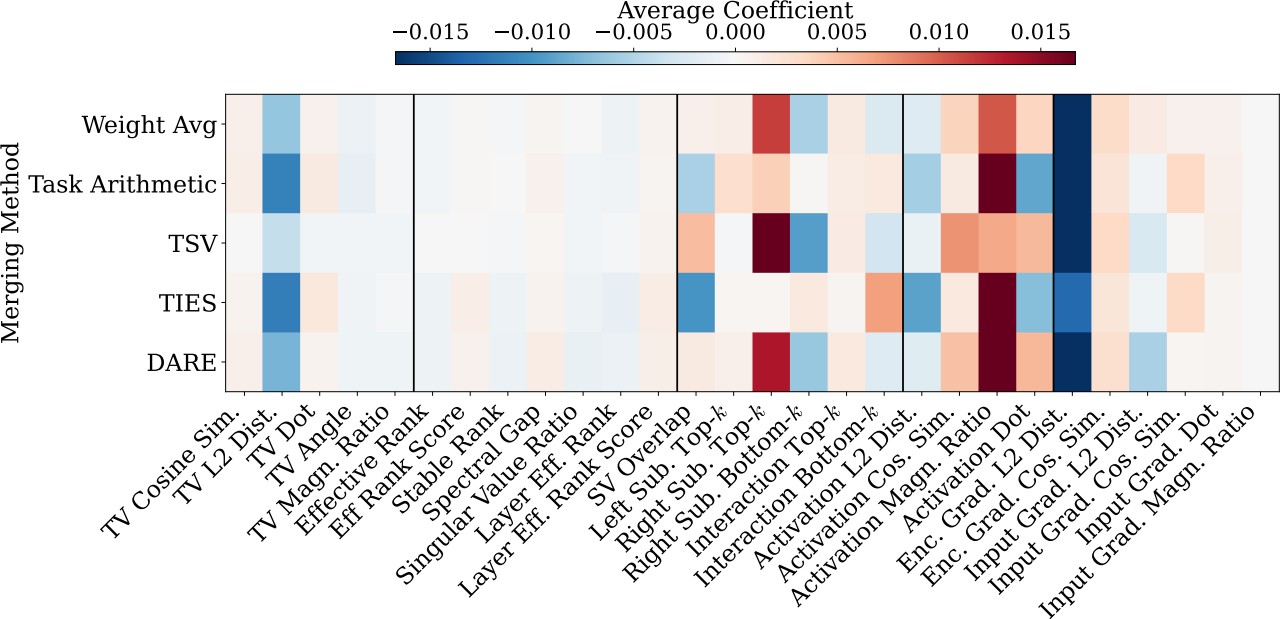

*Figure 5.* Per-method learned coefficients for all 28 metrics, averaged across 20 LOTO folds. **ViT-B/32** backbone.

*Table 7.* Coefficient sign agreement between methods (ViT-B/16). Average cross-method agreement is 0.79.

| Method | TA | WA | TSV | TIES | DARE |
|--------|------|------|------|------|------|
| TA | 1.00 | 0.89 | 0.71 | 0.86 | 0.89 |
| WA | | 1.00 | 0.82 | 0.75 | 0.93 |
| TSV | | | 1.00 | 0.57 | 0.75 |
| TIES | | | | 1.00 | 0.75 |
| DARE | | | | | 1.00 |

**Stable Metrics.** Three metrics exhibit consistent sign and meaningful magnitude across all five merging methods, suggesting they capture fundamental aspects of mergeability:

- **Negative coefficients (higher values predict worse merging):**
  - Encoder gradient $L_2$ distance (avg: $-0.021$, range: $[-0.026, -0.013]$)
  - Input gradient $L_2$ distance (avg: $-0.018$, range: $[-0.020, -0.015]$)
- **Positive coefficients (higher values predict better merging):**
  - Right subspace overlap top-$k$ (avg: $+0.007$, range: $[+0.000, +0.014]$)

**Key Observations.**

- **Gradient-based metrics** consistently receive the largest-magnitude coefficients. The $L_2$ distance metrics for encoder and input gradients are negative across all methods, indicating that large gradient discrepancies between task models are detrimental to merging.
- **Subspace overlap metrics** tend to receive positive coefficients, suggesting that task vectors sharing similar principal directions merge more effectively.
- **Method-specific patterns**: TIES and TA form one cluster; WA, TSV, and DARE form another. The two clusters share few top-5 metrics but converge at top-10.
- **Improved coefficient stability**: L1 regularization yields more stable non-zero coefficients compared to the unconstrained setup, with standard deviations comparable to or smaller than the coefficient means for the dominant metrics

(e.g., encoder gradient $L_2$: $-0.021 \pm 0.010$).

*Table 8.* Average learned coefficients ($\pm$ standard deviation) for each mergeability metric across merging methods, obtained via L1-regularized leave-one-task-out cross-validation ($\lambda = 1.0$, ViT-B/16). Positive coefficients indicate that higher metric values predict better post-merge performance; negative indicate the opposite. Metrics are min-max normalized to $[-1, 1]$.

| | Metric | Task Arithmetic | Weight Averaging | TSV | TIES | DARE |
|---|---|---|---|---|---|---|
| **Task Vector** | TV Cosine Sim | $+0.000$ | $+0.000$ | $-0.000$ | $+0.000$ | $+0.000$ |
| | TV $L_2$ Dist | $-0.007 \pm 0.006$ | $-0.002 \pm 0.003$ | $-0.002 \pm 0.004$ | $-0.017 \pm 0.012$ | $-0.002 \pm 0.003$ |
| | TV Dot Prod | $-0.000$ | $-0.000$ | $-0.000$ | $+0.000$ | $-0.000$ |
| | Weight Angle | $-0.001 \pm 0.001$ | $-0.001 \pm 0.001$ | $-0.001 \pm 0.001$ | $-0.001 \pm 0.001$ | $-0.001 \pm 0.001$ |
| | TV Mag Ratio | $-0.000$ | $-0.000$ | $+0.000$ | $-0.000$ | $-0.000$ |
| **Effective Rank** | Eff Rank | $-0.000$ | $-0.000$ | $-0.000$ | $-0.000$ | $-0.000$ |
| | Eff Rank Score | $+0.000$ | $+0.000$ | $-0.000$ | $+0.000$ | $+0.000$ |
| | Stable Rank | $-0.000$ | $-0.000$ | $-0.000$ | $-0.000$ | $-0.000$ |
| | Spectral Gap | $+0.000$ | $+0.000$ | $-0.000$ | $+0.000$ | $+0.000$ |
| | SV Ratio | $-0.000$ | $-0.000$ | $-0.000$ | $-0.000$ | $+0.000$ |
| | Layer Eff Rank | $-0.000$ | $-0.000$ | $-0.000$ | $-0.000$ | $-0.000$ |
| | Layer Eff Rank Score | $+0.000$ | $+0.000$ | $+0.000$ | $+0.000$ | $+0.000$ |
| **Sub. Overlap** | SV Overlap | $-0.007 \pm 0.003$ | $+0.002 \pm 0.003$ | $+0.007 \pm 0.004$ | $-0.014 \pm 0.009$ | $+0.002 \pm 0.003$ |
| | Left Sub Top-$k$ | $+0.002 \pm 0.003$ | $+0.001 \pm 0.002$ | $-0.001 \pm 0.002$ | $+0.002 \pm 0.003$ | $+0.001 \pm 0.002$ |
| | Right Sub Top-$k$ | $+0.003 \pm 0.004$ | $+0.009 \pm 0.004$ | $+0.014 \pm 0.005$ | $+0.000 \pm 0.003$ | $+0.008 \pm 0.005$ |
| | Right Sub Bot-$k$ | $+0.001 \pm 0.002$ | $-0.007 \pm 0.004$ | $-0.015 \pm 0.004$ | $+0.007 \pm 0.004$ | $-0.008 \pm 0.004$ |
| | Interact Top-$k$ | $+0.001 \pm 0.001$ | $+0.001 \pm 0.001$ | $+0.001 \pm 0.001$ | $+0.001 \pm 0.001$ | $+0.001 \pm 0.001$ |
| | Interact Bot-$k$ | $+0.006 \pm 0.004$ | $-0.001 \pm 0.001$ | $-0.000$ | $+0.015 \pm 0.010$ | $+0.001 \pm 0.001$ |
| **Activation** | Act $L_2$ Dist | $-0.001 \pm 0.001$ | $-0.002 \pm 0.002$ | $-0.002 \pm 0.002$ | $+0.000$ | $-0.001 \pm 0.001$ |
| | Act Cosine Sim | $+0.001 \pm 0.002$ | $+0.007 \pm 0.003$ | $+0.010 \pm 0.005$ | $-0.000$ | $+0.007 \pm 0.003$ |
| | Act Mag Ratio | $-0.001 \pm 0.001$ | $-0.001 \pm 0.001$ | $-0.001 \pm 0.001$ | $-0.001 \pm 0.001$ | $-0.001 \pm 0.001$ |
| | Act Dot Prod | $+0.001 \pm 0.002$ | $+0.009 \pm 0.005$ | $+0.011 \pm 0.006$ | $-0.001 \pm 0.001$ | $+0.004 \pm 0.004$ |
| **Gradient** | Enc Grad Cos | $+0.001 \pm 0.001$ | $+0.001 \pm 0.001$ | $+0.001 \pm 0.001$ | $+0.001 \pm 0.001$ | $+0.001 \pm 0.001$ |
| | Enc Grad $L_2$ | $-0.016 \pm 0.010$ | $-0.025 \pm 0.011$ | $-0.026 \pm 0.008$ | $-0.013 \pm 0.008$ | $-0.026 \pm 0.010$ |
| | Enc Grad Dot | $+0.001 \pm 0.001$ | $+0.001 \pm 0.001$ | $+0.001 \pm 0.001$ | $+0.001 \pm 0.001$ | $+0.001 \pm 0.001$ |
| | Input Grad Cos | $+0.001 \pm 0.001$ | $+0.001 \pm 0.001$ | $+0.001 \pm 0.001$ | $+0.001 \pm 0.001$ | $+0.001 \pm 0.001$ |
| | Input Grad $L_2$ | $-0.018 \pm 0.010$ | $-0.020 \pm 0.014$ | $-0.019 \pm 0.012$ | $-0.015 \pm 0.006$ | $-0.019 \pm 0.011$ |
| | Input Grad Dot | $-0.001 \pm 0.001$ | $-0.001 \pm 0.001$ | $-0.001 \pm 0.001$ | $-0.001 \pm 0.001$ | $-0.001 \pm 0.001$ |

## A.4. Mergeability Metrics: Definitions and Implementation Details

This appendix provides formal definitions and implementation details for the mergeability metrics summarized in Section 3 of the main paper.

We introduce a suite of 28 pairwise metrics to study model compatibility. Since single metrics in isolation exhibit limited predictive power (A.2), we leverage their combined expressivity to characterize mergeability. These metrics are computed without performing the merge, enabling efficient pre-screening of model pairs. For layer-wise metrics, we report the mean across all layers. We categorize these metrics into five distinct groups, detailed below and outlined in Table 9.

### A.4.1. TASK VECTOR GEOMETRY METRICS

Given two task vectors $\boldsymbol{\tau}_A, \boldsymbol{\tau}_B \in \mathbb{R}^D$, defined as the flattened 1D differences $\boldsymbol{\theta}_i - \boldsymbol{\theta}_0$ between fine-tuned and pretrained weights, the most direct approach to measuring compatibility examines their geometric relationship in weight space. The intuition is that task vectors pointing in similar directions may interfere less during merging.

- **Cosine Similarity**: Measures directional alignment between task vectors: $\frac{\tau_A \cdot \tau_B}{\|\tau_A\|_2 \|\tau_B\|_2}$.
- $L_2$ **Distance**: Euclidean distance between task vectors: $\|\tau_A - \tau_B\|_2$. Small distances indicate similar updates.

*Table 9.* Summary of mergeability metrics by category.

| Category | Metrics | Count |
|---|---|---|
| Task Vector | cos sim, $L_2$ dist., dot, angle, magnitude ratio | 5 |
| Effective Rank | eff rank, stable rank, spectral gap, etc. | 7 |
| Subspace Overlap | SV overlap, subspace overlap, interaction | 6 |
| Activation-Based | $L_2$ dist., cosine sim., magnitude ratio, dot | 4 |
| Gradient-Based | encoder/input grad. $L_2$ dist., cos sim, dot | 6 |
| **Total** | | **28** |

- **Dot Product**: Captures both direction and magnitude: $\tau_A \cdot \tau_B$. This potentially identifies cases where one task dominates the weight space.
- **Weight Space Angle**: The angle $\theta = \arccos(\text{cos sim})$ in degrees, providing an interpretable measure of directional difference.
- **Magnitude Ratio**: Expressed as the ratio between the two task vector magnitudes: $\min(\|\tau_A\|, \|\tau_B\|)/\max(\|\tau_A\|, \|\tau_B\|)$. A ratio close to 1 suggests balanced contributions from both tasks

### A.4.2. EFFECTIVE RANK METRICS

Recent work shows task arithmetic benefits from low-rank task vectors (Liu et al., 2025). We propose metrics based on the *effective rank* of task vectors to capture this property.

Given the $2 \times D$ matrix $\mathbf{T} = [\tau_A; \tau_B]^\top$ formed by stacking the two task vectors, we compute its singular value decomposition to obtain singular values $\sigma_1 \geq \sigma_2$.

- **Effective Rank**: The participation ratio based on singular value entropy:

$$\text{EffRank}(\mathbf{T}) = \exp\left(-\sum_i p_i \log p_i\right), \quad p_i = \frac{\sigma_i}{\sum_j \sigma_j} \tag{4}$$

An effective rank of 1 indicates a perfect low-rank structure (all mass in one direction), while higher values indicate more diffuse subspaces.

- **Effective Rank Mergeability Score**: A normalized version mapping effective rank from $[1, 2]$ to $[1, 0]$, where higher scores indicate better compatibility.
- **Stable Rank**: An alternative dimensionality measure using the squared Frobenius norm:

$$\text{StableRank}(\mathbf{T}) = \frac{(\sum_i \sigma_i)^2}{\sum_i \sigma_i^2} \tag{5}$$

- **Spectral Gap**: The normalized difference between singular values $(\sigma_1 - \sigma_2)/\sigma_1$. A large gap indicates one dominant direction, suggesting alignment.
- **Singular Value Ratio**: The ratio $\sigma_2/\sigma_1$. Smaller values indicate stronger alignment.
- **Layer-wise Effective Rank**: Computes effective rank per layer and averages, weighted by layer update magnitude $\|\tau_A^{(\ell)}\| + \|\tau_B^{(\ell)}\|$. This provides finer-grained insight than global effective rank.
- **Layer-wise Effective Rank Mergeability Score**: The normalized version of layer-wise effective rank.

### A.4.3. SUBSPACE OVERLAP METRICS

These metrics analyze how the principal directions modified by each task vector relate to each other, based on the Singular Value Decomposition (SVD) of individual weight matrices, which decomposes a matrix $W \in \mathbb{R}^{m \times n}$ as:

$$W = U\Sigma V^\top \tag{6}$$

where $U \in \mathbb{R}^{m \times m}$ and $V \in \mathbb{R}^{n \times n}$ are orthogonal matrices containing the left and right singular vectors, and $\Sigma \in \mathbb{R}^{m \times n}$ is a diagonal matrix containing the singular values $\sigma_1 \geq \sigma_2 \geq \cdots \geq 0$.

- **Singular Value Overlap**: For each 2D task vector matrix, computes the cosine similarity between the top-100 normalized singular value distributions, averaged across all layers.

- **Left Subspace Overlap**: Measures overlap between the top-$k$ left singular vectors (columns of $\mathbf{U}$) across weight matrices using the Frobenius norm of $\mathbf{U}_A^{(k)\top}\mathbf{U}_B^{(k)}$. The value of $k$ is set to 10 here and in the following metrics.

- **Right Subspace Overlap (Top-$k$ and Bottom-$k$)**: Similarly measures overlap between right singular vectors (rows of $\mathbf{V}$). We compute this for both the strongest (top-$k$) and weakest (bottom-$k$) singular directions, as different merging methods may be sensitive to different spectral components.

- **Interaction Matrix Overlap (Top-$k$ and Bottom-$k$)**: Computes the interaction matrix $\mathbf{M} = \mathbf{V}_A^{(k)\top}\mathbf{V}_B^{(k)}$ whose singular values represent cosines of principal angles between subspaces. We report the mean squared singular value of $\mathbf{M}$.

### A.4.4. ACTIVATION-BASED METRICS

While weight-space metrics are computationally efficient, they may miss functional similarities. We complement them with activation-based metrics that measure how similarly the fine-tuned models process inputs.

Given a small calibration set $\mathcal{D}_{\mathrm{cal}}$ (10 random samples per dataset in our experiments), we extract activations from the final encoder layer of both models and compute:

- **Activation $L_2$ Distance**: $\|\bar{\mathbf{a}}_A - \bar{\mathbf{a}}_B\|_2$ where $\bar{\mathbf{a}}$ denotes the mean activation over calibration samples.
- **Activation Cosine Similarity**: Directional alignment of mean activations.
- **Activation Magnitude Ratio**: Ratio of activation norms, indicating whether scales are similar.
- **Activation Dot Product**: Captures both direction and magnitude of activation alignment.

### A.4.5. GRADIENT-BASED METRICS

Gradient alignment has been shown to correlate with multi-task learning success (Yu et al., 2020). We leverage the same calibration set as in activation-based metrics, and measure gradient similarity at two levels:

**Encoder Gradient Metrics.** We compute gradients of the cross-entropy loss with respect to encoder parameters:

- **Encoder Gradient Cosine Similarity**: Measures the alignment of the directions in which models require updates: $\frac{g_A \cdot g_B}{\|g_A\|_2\|g_B\|_2}$.
- **Encoder Gradient $L_2$ Distance**: Measures the Euclidean magnitude of the difference between gradient vectors: $\|g_A - g_B\|_2$.
- **Encoder Gradient Dot Product**: Captures both the direction and magnitude of gradient agreement: $g_A \cdot g_B$.

**Input Gradient Metrics.** We compute gradients with respect to input images to capture how models attend to them:

- **Input Gradient Cosine Similarity**: Measures whether models focus on similar input regions.
- **Input Gradient $L_2$ Distance**: Magnitude of input sensitivity difference.
- **Input Gradient Dot Product**: Combined direction and magnitude measure.

Table 9 summarizes all 28 metrics and their computational requirements. Weight-based and effective rank metrics require only the task vectors and are thus the most efficient. Activation and gradient metrics require a small calibration set and forward/backward passes, but capture functional properties that pure weight-space metrics may miss.

### A.5. Computational Costs of Metrics

In this appendix, we analyze the computational cost of each metric category and relate it to predictive value, providing guidance for practitioners on which metrics offer the best cost-benefit trade-off.

### A.5.1. METRIC CATEGORIES AND COMPUTATIONAL REQUIREMENTS

Table 10 summarizes the computational requirements for each metric category. We categorize costs based on the primary computational operation required.

*Table 10.* Computational cost analysis of mergeability metric categories. Performance impact shows the change in validation correlation ($\Delta r$) when the category is excluded from L1-regularized LOTO.

| Category | Primary Operation | Cost | Data Required | $\Delta r$ |
|---|---|---|---|---|
| Gradient-based | Forward + Backward | High | Train data | $-0.167$ |
| Activation | Forward only | Medium | Train data | $-0.051$ |
| Subspace | SVD decomposition | Medium | Weights only | $-0.045$ |
| Task Vector | Weight arithmetic | Low | Weights only | $-0.032$ |
| Effective Rank | SVD decomposition | Medium | Weights only | $+0.010$ |

### A.5.2. DETAILED COST BREAKDOWN

**Gradient-based Metrics (High Cost)**   These metrics require computing gradients with respect to either the encoder parameters or input images. For each model pair:

- Forward pass through both models on shared validation data
- Backward pass to compute gradients
- Similarity computation between gradient vectors

The backward pass roughly doubles the compute compared to forward-only approaches. However, gradient-based metrics show the largest predictive impact ($-0.167$ when excluded), making them essential despite their cost.

**Activation Metrics (Medium Cost)**   Activation-based metrics require:

- Forward pass through both models on shared data
- Storage and comparison of intermediate activations

Memory requirements scale with batch size and model depth. These metrics provide moderate predictive value ($-0.051$ when excluded) at lower cost than gradient-based approaches.

**Subspace Metrics (Medium Cost)**   Subspace overlap metrics require:

- SVD decomposition of weight matrices (or task vectors)
- Computation of subspace overlap/similarity measures

SVD complexity is $O(\min(m, n) \cdot mn)$ for an $m \times n$ matrix. These metrics are data-free (require only model weights) but computationally non-trivial. Predictive value is moderate ($-0.045$ when excluded).

**Task Vector Metrics (Low Cost)**   Task vector metrics involve:

- Subtraction of pretrained weights from fine-tuned weights
- Vector similarity computations (cosine similarity, L2 distance)

These are the cheapest metrics to compute, requiring only weight arithmetic. Despite low cost, they provide meaningful signal ($-0.032$ when excluded).

**Effective Rank Metrics (Medium Cost – Not Recommended)**   Effective rank metrics require SVD decomposition similar to subspace metrics. However, our ablation study shows that excluding these metrics *improves* performance ($+0.010$), indicating they introduce noise rather than signal. We recommend **not computing these metrics** for mergeability prediction.

## A.6. Optimization Objective Comparison: MSE vs. Correlation

A critical design choice in learning mergeability prediction coefficients is the optimization objective. We compare two objectives: maximizing Pearson correlation versus minimizing mean squared error (MSE).

**Objective Functions.** For Pearson correlation maximization, we optimize:

$$\mathcal{L}_{\text{Pearson}} = -\frac{\sum_i (y_i - \bar{y})(\hat{y}_i - \bar{\hat{y}})}{\sqrt{\sum_i (y_i - \bar{y})^2 \sum_i (\hat{y}_i - \bar{\hat{y}})^2}} + \lambda \|\mathbf{w}\|_1 \tag{7}$$

where $y_i$ is the actual merge performance, $\hat{y}_i = \mathbf{w}^\top \mathbf{x}_i$ is the predicted mergeability, and $\mathbf{w}$ are the learned coefficients.

For MSE minimization:

$$\mathcal{L}_{\text{MSE}} = \frac{1}{N} \sum_i (y_i - \hat{y}_i)^2 \tag{8}$$

**Results.** Tables 11 and 12 present the validation Pearson correlation for each objective across all merging methods on ViT-B-16 and ViT-B-32.

*Table 11.* ViT-B-16: Validation Pearson correlation ($r$) for different optimization objectives. Higher is better.

| Method | Pearson + L1 | MSE | Difference |
|---|---|---|---|
| Weight Avg | 0.589 | 0.244 | +0.345 |
| Arithmetic | 0.445 | 0.061 | +0.384 |
| TSV | 0.627 | 0.187 | +0.440 |
| TIES | 0.463 | 0.098 | +0.365 |
| DARE | 0.596 | 0.143 | +0.453 |
| **Average** | **0.544** | 0.147 | +0.397 |

*Table 12.* ViT-B-32: Validation Pearson correlation ($r$) for different optimization objectives. Higher is better.

| Method | Pearson + L1 | MSE | Difference |
|---|---|---|---|
| Weight Avg | 0.699 | 0.047 | +0.652 |
| Arithmetic | 0.619 | 0.131 | +0.488 |
| TSV | 0.692 | 0.153 | +0.539 |
| TIES | 0.543 | 0.147 | +0.396 |
| DARE | 0.684 | 0.236 | +0.448 |
| **Average** | **0.647** | 0.143 | +0.505 |

**Analysis.   Pearson correlation substantially outperforms MSE across all architectures.** For ViT-B-16, the average validation correlation is 0.544 with Pearson versus 0.147 with MSE ($3.7\times$ improvement). For ViT-B-32, the gap is 0.648 versus 0.143 ($4.5\times$).

This performance gap arises from the fundamental mismatch between MSE and our prediction goal. Our objective is to *rank* task pairs by mergeability, predicting whether pair A will merge better than pair B, rather than predicting exact merge performance values. Correlation-based objectives directly optimize for ranking quality: they are invariant to the scale and shift of predictions, focusing entirely on whether the relative ordering is preserved.

In contrast, MSE optimization must simultaneously learn (1) the correct ranking and (2) the correct output scale. Since merge performance values are bounded between 0 and 1 (normalized accuracy), while our linear predictions $\hat{y} = \mathbf{w}^\top \mathbf{x}$ can take arbitrary values, the optimization must implicitly learn to constrain outputs to a valid range. This additional burden degrades ranking performance, and worsens with fewer training samples.

**Conclusion.** We adopt Pearson correlation with L1 regularization as our default objective. The finding holds consistently across two architectures, confirming that correlation-based optimization is the right inductive bias for mergeability prediction.

## A.7. Linear vs. Neural Network Predictors

A natural question arises when designing mergeability predictors: can neural networks capture non-linear relationships between metrics and merge performance that linear models miss? To investigate this, we compare our L1-regularized linear approach against Multi-Layer Perceptrons (MLPs) trained separately for each merge method.

**Experimental Setup.** We train separate MLPs for each merge method using the same Leave-One-Task-Out (LOTO) cross-validation protocol. Each MLP consists of a single hidden layer with 8 units, ReLU activation, and dropout regularization (p=0.4). Models are trained for 300 epochs with early stopping based on validation loss. We evaluate both approaches on models fine-tuned with the AdamW optimizer.

**Results.** Table 13 presents the aggregate validation Pearson correlation for both approaches across all merge methods. The L1-regularized linear model consistently outperforms the MLP predictor.

*Table 13.* Comparison of $L_1$-regularized linear models vs. MLPs for mergeability prediction. Values represent aggregate validation Pearson correlation ($r$) under LOTO cross-validation. Higher is better.

| Method | MLP | $L_1$ **Linear** |
|---|---|---|
| Weight Avg | 0.590 | 0.590 |
| Task Arithmetic | 0.335 | 0.445 |
| TSV | 0.584 | 0.627 |
| TIES | 0.452 | 0.463 |
| DARE | 0.502 | 0.596 |
| **Average** | 0.493 | **0.544** |

The L1 linear approach achieves a mean validation correlation of $0.544$ compared to $0.493$ for the MLP, representing a $10\%$ relative improvement. Notably, the L1 linear model wins over all 5 merge methods.

**Why Linear Models Suffice.** Several factors explain the superior performance of linear models in this setting:

1. **Limited training data.** With approximately 170 model pairs per fold, the effective training set is small by deep learning standards. MLPs with even modest capacity risk overfitting, as evidenced by the high variance in per-fold results (standard deviation of 0.25–0.31 across folds).
2. **Inherently linear relationships.** The mergeability metrics we consider—gradient similarities, subspace overlaps, and task vector distances—capture fundamental geometric relationships between model weight spaces. These relationships translate to merge performance through approximately linear mappings.
3. **L1 regularization as implicit feature selection.** The L1 penalty encourages sparse solutions, effectively performing feature selection during training. This prevents the model from fitting to noise in less informative metrics, a form of regularization that the MLP's dropout alone cannot replicate.

**Interpretability Advantage.** Beyond predictive performance, linear models offer a crucial interpretability advantage. The learned coefficients directly indicate each metric's contribution to predicting merge success, enabling practitioners to understand *why* certain model pairs are predicted to merge well. In contrast, MLP weights lack such transparency, making it difficult to extract actionable insights about which properties of fine-tuned models drive mergeability.

**Conclusion.** Our results demonstrate that for mergeability prediction, simplicity prevails. L1-regularized linear models not only achieve superior generalization but also provide interpretable coefficients that reveal the relative importance of different metrics. This finding aligns with the broader machine learning principle that model complexity should match the structure of the underlying problem and the available data. For mergeability prediction, where training examples are inherently limited and interpretability is valuable, linear models represent the right inductive bias.

### A.8. Robustness to Random Initialization

To verify that our L1-regularized LOTO framework produces stable results regardless of random initialization, we run the optimization with 10 different random seeds on ViT-B-16 and measure the variance in validation performance and metric selection.

### A.9. Validation Performance Stability

Table 14 shows the mean, standard deviation, and range of validation Pearson correlation across 10 random seeds for each merging method.

*Table 14.* Robustness of L1 LOTO across 10 random seeds (ViT-B-16). Low standard deviations indicate that performance is not sensitive to random initialization.

| Method | Val $r$ (mean) | Val $r$ (std) | Range | #Features |
|---|---|---|---|---|
| Weight Avg | 0.609 | 0.018 | [0.577, 0.632] | 16.8 |
| Arithmetic | 0.466 | 0.017 | [0.441, 0.489] | 17.0 |
| TSV | 0.639 | 0.022 | [0.596, 0.666] | 16.3 |
| TIES | 0.469 | 0.011 | [0.450, 0.482] | 16.1 |
| DARE | 0.622 | 0.010 | [0.601, 0.638] | 16.7 |
| **Average** | 0.561 | 0.016 | — | 16.6 |

**Key Findings.**

- **Low variance across seeds**: The standard deviation of validation correlation ranges from 0.010 (DARE) to 0.022 (TSV), indicating that random initialization has minimal impact on final performance.

- **Consistent feature selection**: The number of selected features is stable across seeds (16.1–17.0 on average), suggesting that the L1 regularization consistently identifies a similar-sized subset of predictive metrics.

- **Narrow performance range**: For all methods, the difference between best and worst seed is less than 0.07 in validation correlation, demonstrating robust convergence.

### A.10. Metric Selection Stability

Beyond overall performance, we verify that the same metrics are selected across different random seeds. Table 15 shows that `encoder_gradient_l2_distance` is selected with 100% frequency across all seeds and all mergers, confirming its role as a universal predictor.

*Table 15.* Selection frequency and coefficient stability for `encoder_gradient_l2_distance` across 10 random seeds.

| Method | Selection Freq | Coef (mean) | Coef (std) |
|---|---|---|---|
| Weight Avg | 100% (all seeds) | $-0.026$ | 0.002 |
| Arithmetic | 100% (all seeds) | $-0.016$ | 0.001 |
| TSV | 100% (all seeds) | $-0.022$ | 0.002 |
| TIES | 100% (all seeds) | $-0.012$ | 0.001 |
| DARE | 100% (all seeds) | $-0.025$ | 0.002 |

The coefficient values are also stable across seeds, with standard deviations of only 0.001–0.002. This indicates that the learned relationship between gradient distance and mergeability is consistent regardless of optimization initialization.

### A.11. Implications

These robustness results demonstrate that:

1. The L1 LOTO framework produces reliable, reproducible predictions

2. Users can run the optimization once without needing ensemble methods or multiple restarts

3. The identified predictive metrics (particularly gradient-based) are genuine signals, not artifacts of specific random initializations

## A.12. Gradient Magnitude Regularization for Better Model Merging

### A.12.1. MOTIVATION

Our analysis of metric contributions reveals that gradient-based metrics, specifically `encoder_gradient_l2_distance` and `input_gradient_l2_distance`, are the most consistently selected predictors of mergeability across merging methods. These metrics measure how similarly two fine-tuned models respond to input perturbations, capturing functional similarity beyond weight-space comparisons. This finding motivates a regularization strategy that directly encourages gradient proximity during fine-tuning.

### A.12.2. METHOD

Gradient magnitude regularization introduces a penalty on the norm of the gradient at each optimization step:

$$\mathcal{L}_{\text{total}} = \mathcal{L}_{\text{CE}} + \lambda \sum_{l} \|\nabla_{\theta^{(l)}} \mathcal{L}_{\text{CE}}\|_2^2 \tag{9}$$

where $\mathcal{L}_{\text{CE}}$ is the cross-entropy loss, $\theta^{(l)}$ are the parameters of layer $l$, and $\lambda$ controls the regularization strength. By penalizing large gradients, this regularization encourages smoother optimization trajectories with smaller parameter updates. Implicitly, this brings different fine-tuned models closer together in gradient space: models trained with constrained gradients will exhibit more similar gradient responses to shared inputs, reducing the gradient-based distance metrics that our analysis identifies as key mergeability predictors.

### A.12.3. EXPERIMENTAL SETUP

We evaluate gradient magnitude regularization ($\lambda = 1$) on pairwise model merging using the ViT-B-16 architecture. Pairwise merging combines exactly two fine-tuned models at a time, providing a controlled environment to study merging dynamics. We evaluate all 190 unique pairs from the N20 benchmark (20 datasets), computing the average normalized accuracy across both tasks for each pair.

### A.12.4. RESULTS

Table 16 presents the average normalized accuracy across all 190 pairwise merges for five merging methods.

*Table 16.* Effect of gradient magnitude regularization on pairwise merging (N20 benchmark, 190 pairs). $\Delta$ indicates absolute percentage point change from baseline.

| Merger | Baseline | Grad Mag ($\lambda$=1) | $\Delta$ |
|---|---|---|---|
| Arithmetic | 0.9102 | 0.9174 | +0.72% |
| DARE | 0.9563 | 0.9628 | +0.65% |
| TSV | 0.9818 | 0.9857 | +0.39% |
| TIES | 0.8415 | 0.8404 | −0.11% |
| Weight Avg | 0.9586 | 0.9625 | +0.39% |
| **Average** | **0.9297** | **0.9338** | **+0.41%** |

### A.12.5. ANALYSIS

Gradient magnitude regularization improves pairwise merging performance for four out of five merging methods, with an average improvement of +0.41% in normalized accuracy.

**Consistent Improvements for Most Mergers.** Arithmetic (+0.72%), DARE (+0.65%), TSV (+0.39%), and Weight Averaging (+0.39%) all benefit from gradient magnitude regularization. This is consistent with our metric importance

analysis, which identified gradient-based metrics as the top predictors of mergeability for these methods. By regularizing gradient magnitudes during fine-tuning, we reduce the gradient distance between models, directly optimizing the metrics that most strongly predict successful merging.

**Why TIES Does Not Improve.** TIES is the only merging method that exhibits slight performance degradation ($-0.11\%$) with gradient magnitude regularization. This exception is explained by our metric importance analysis: unlike the other four mergers, gradient metrics are *not* among the most dominant predictors for TIES. Table 17 shows the top-5 metrics by coefficient magnitude for each merger. For Weight Averaging, Arithmetic, TSV, and DARE, gradient metrics (`encoder_gradient_l2_distance` and `input_gradient_l2_distance`) occupy positions #1 and #2. In contrast, TIES has `interaction_matrix_overlap_bottom_k` and `task_vector_l2_distance` as its top predictors, with gradient metrics appearing only at positions #3 and #5. Consequently, a regularization strategy designed to improve gradient proximity does not address the factors most relevant to TIES performance.

*Table 17.* Top-5 metrics by L1 LOTO coefficient magnitude for each merger. Gradient metrics (bold) dominate for all mergers except TIES.

| Merger | #1 | #2 | #3 | #4 | #5 |
|---|---|---|---|---|---|
| Weight Avg | **enc_grad** | **inp_grad** | act_cos | rs_top | rs_bot |
| Arithmetic | **inp_grad** | **enc_grad** | tv_l2 | im_bot | sv_overlap |
| TSV | **enc_grad** | **inp_grad** | rs_bot | rs_top | act_cos |
| TIES | im_bot | tv_l2 | **inp_grad** | sv_overlap | **enc_grad** |
| DARE | **enc_grad** | **inp_grad** | act_cos | rs_top | rs_bot |

**Quantifying the Gradient Contribution.** To make fair comparisons across mergers, we normalize the absolute coefficient values to sum to 1, yielding the relative contribution of each metric. Table 18 shows the combined contribution of gradient metrics for each merger. For Weight Averaging, Arithmetic, TSV, and DARE, all six gradient metrics collectively account for 43–54% of the total predictive signal. In contrast, TIES allocates only 31% to gradient metrics, instead relying more heavily on `interaction_matrix_overlap_bottom_k` (16.8%) and `task_vector_l2_distance` (16.4%). This quantitative difference explains why gradient magnitude regularization improves the former group but not TIES: the regularization targets roughly half of the predictive signal for most mergers, but less than one-third for TIES.

*Table 18.* Normalized contribution of gradient metrics to mergeability prediction (ViT-B-16). Values represent the fraction of total predictive weight allocated to each metric after normalizing absolute coefficients to sum to 1. Total covers all six gradient-based metrics.

| Merger | enc_grad_l2 | inp_grad_l2 | Total (all 6) |
|---|---|---|---|
| Weight Avg | 24.8% | 19.3% | **49.8%** |
| Arithmetic | 18.4% | 19.8% | **43.5%** |
| TSV | 23.0% | 17.4% | **43.3%** |
| TIES | 13.1% | 15.3% | 30.9% |
| DARE | 27.9% | 20.6% | **54.4%** |

This finding validates our metric-driven approach to regularization design: regularization strategies should target the metrics that are most predictive for the specific merging method being used. A universal regularization strategy may not benefit all mergers equally, and the effectiveness of a particular regularization can be predicted by examining which metrics dominate the mergeability prediction for that method.

### A.12.6. SUMMARY

Gradient magnitude regularization provides consistent improvements for merging methods where gradient-based metrics are the dominant mergeability predictors (Arithmetic, DARE, TSV, Weight Averaging). The exception of TIES, where gradient metrics are not the top predictors, confirms that effective regularization strategies should be informed by metric importance analysis. This demonstrates a practical application of our mergeability prediction framework: the learned metric coefficients not only predict merging success but also guide the design of regularization strategies for improved fine-tuning.

*Table 19.* LOTO cross-validation results comparing $L_1$-regularized ($\lambda = 1.0$, left) and unregularized ($\lambda = 0$, right) linear predictors. Statistics show mean $\pm$ std of Pearson $r$ across folds.

| Method | **L1 ($\lambda = 1.0$)** | | **No L1 ($\lambda = 0$)** | |
|---|---|---|---|---|
| | Train $r$ | Val $r$ | Train $r$ | Val $r$ |
| *ViT-B/16 (20 tasks, 20 folds)* | | | | |
| Weight Avg | $0.702 \pm .072$ | $0.589 \pm .175$ | $0.821 \pm .016$ | $0.567 \pm .290$ |
| Task Arithmetic | $0.574 \pm .054$ | $0.445 \pm .163$ | $0.654 \pm .051$ | $0.414 \pm .241$ |
| TSV | $0.754 \pm .026$ | $0.627 \pm .155$ | $0.859 \pm .011$ | $0.639 \pm .247$ |
| TIES | $0.552 \pm .072$ | $0.463 \pm .135$ | $0.648 \pm .039$ | $0.442 \pm .208$ |
| DARE | $0.722 \pm .047$ | $0.596 \pm .169$ | $0.820 \pm .016$ | $0.564 \pm .285$ |
| *ViT-B/32 (20 tasks, 20 folds)* | | | | |
| Weight Avg | $0.799 \pm .048$ | $0.699 \pm .165$ | $0.887 \pm .013$ | $0.689 \pm .228$ |
| Task Arithmetic | $0.739 \pm .049$ | $0.619 \pm .170$ | $0.831 \pm .016$ | $0.663 \pm .201$ |
| TSV | $0.784 \pm .066$ | $0.692 \pm .165$ | $0.885 \pm .015$ | $0.710 \pm .189$ |
| TIES | $0.671 \pm .052$ | $0.543 \pm .150$ | $0.795 \pm .015$ | $0.623 \pm .162$ |
| DARE | $0.801 \pm .034$ | $0.684 \pm .172$ | $0.888 \pm .013$ | $0.694 \pm .227$ |

## A.13. Effect of $L_1$ Regularization

We adopt an $L_1$-penalized linear predictor as our primary setup. The loss is:

$$\mathcal{L} = -\text{Pearson}(\hat{s}, s) + \lambda \sum_{i=1}^{k} |\tilde{w}^i|$$

with $\lambda = 1.0$ (selected without additional tuning). To quantify the benefit of regularization, we compare against an unregularized baseline ($\lambda = 0$), which uses all 28 metrics simultaneously.

Table 19 shows LOTO validation performance under both settings for ViT-B/16 and ViT-B/32. The $L_1$ predictor consistently reduces fold-to-fold variance: the average standard deviation of validation $r$ drops from $0.254$ to $0.159$ on ViT-B/16, and from $0.201$ to $0.164$ on ViT-B/32. On ViT-B/16, regularization also improves average validation correlation across all five methods. On ViT-B/32, the dense baseline achieves marginally higher average validation $r$ ($0.676$ vs. $0.648$), but does so with significantly higher variance and nearly double the number of active features ($27.8$ vs. $18.5$). Overall, $L_1$ regularization achieves comparable or better predictive accuracy using roughly $60\%$ of the features, trading a small amount of average performance on one backbone for substantially more stable, reproducible estimates across folds.

Figure 6 further illustrates the effect of $L_1$ regularization on the learned coefficients. The coefficient heatmaps (top row) show that effective-rank metrics collapse to near-zero under regularization, while subspace overlap and gradient-based metrics retain the highest predictive weight. The non-zero frequency heatmaps (bottom row) confirm this pattern: effective-rank metrics are consistently discarded across folds, while gradient and subspace metrics are selected near $100\%$ of the time on both backbones. Together, these results confirm that the stable predictors identified in the main paper are not artifacts of collinearity or coefficient scaling, but reflect a robust, parsimonious signal of mergeability.

## A.14. Metric Ablation Analysis

To validate the coefficient analysis from Section 6, we perform ablation experiments where entire groups of metrics are excluded from the L1-regularized linear optimization, and measure the change in validation Pearson correlation across both ViT-B/16 and ViT-B/32.

### A.14.1. METRIC CATEGORIES

We organize our 28 metrics into five categories:

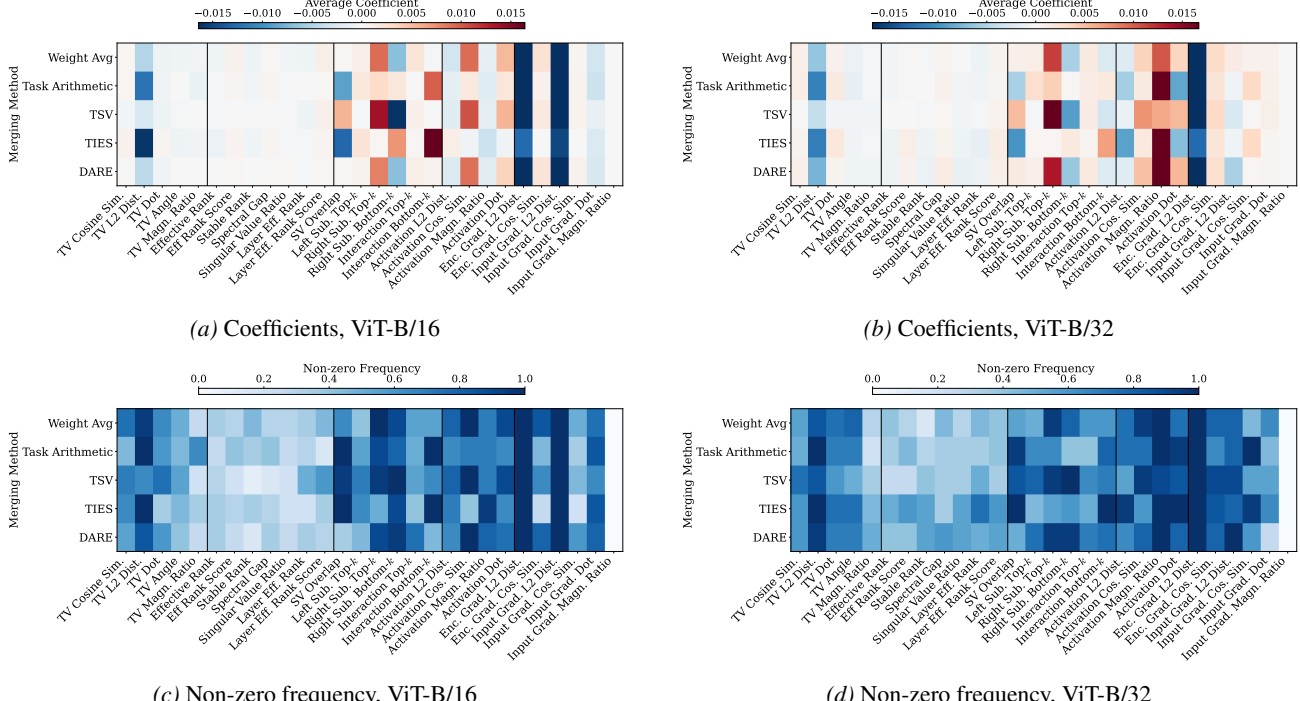

*(a)* Coefficients, ViT-B/16

*(b)* Coefficients, ViT-B/32

*(c)* Non-zero frequency, ViT-B/16

*(d)* Non-zero frequency, ViT-B/32

*Figure 6.* $L_1$-regularized LOTO results for ViT-B/16 and ViT-B/32. Top: average learned coefficients across folds. Bottom: fraction of folds in which each metric's coefficient is non-zero. Vertical lines delimit metric categories.

**Subspace Metrics (6 metrics):** `right_subspace_overlap_top_k`, `right_subspace_overlap_bottom_k`, `subspace_overlap`, `singular_value_overlap`, `interaction_matrix_overlap_top_k`, `interaction_matrix_overlap_bottom_k`

**Gradient-Based Metrics (6 metrics):** `encoder_gradient_{l2_distance, cosine_similarity}`, `input_gradient_{l2_distance, cosine_similarity, dot_product, magnitude_ratio}`

**Effective Rank Metrics (7 metrics):** `effective_rank`, `effective_rank_mergeability_score`, `layerwise_effective_rank`, `layerwise_effective_rank_mergeability_score`, `stable_rank`, `spectral_gap`, `singular_value_ratio`

**Task Vector Metrics (5 metrics):** `task_vector_cosine_similarity`, `task_vector_l2_distance`, `task_vector_dot_product`, `task_vector_magnitude_ratio`, `weight_space_angle`

**Activation Metrics (4 metrics):** `activation_l2_distance`, `activation_cosine_similarity`, `activation_magnitude_ratio`, `activation_dot_product`

### A.14.2. RESULTS

Table 20 reports the validation Pearson correlation for each ablation on both backbones.

### A.14.3. ANALYSIS

**Gradient Metrics Are the Most Critical Category.** Removing gradient-based metrics causes the largest and most consistent performance drop across both backbones and all five merging methods (average $\Delta r = -0.149$ on ViT-B/16, $-0.131$ on ViT-B/32). The effect is largest for Weight Averaging and DARE (both $> -0.15$ on each backbone) and smallest for TIES. This confirms the stable-metric analysis: `encoder_gradient_l2_distance` and `encoder_gradient_cosine_similarity` consistently carry the highest-magnitude coefficients across all meth-

*Table 20.* Full metric category ablation results. Validation Pearson correlation ($r$) reported as mean across LOTO folds; $\Delta$ indicates change from the full 28-metric baseline (in parentheses).

| Method | Baseline (28) | No Subspace (−6) | No Gradient (−6) | No EffRank (−7) | No TaskVec (−5) | No Activ (−4) |
|---|---|---|---|---|---|---|
| *ViT-B/16* | | | | | | |
| Weight Avg | 0.589 | 0.599 (+0.010) | 0.382 (−0.207) | 0.622 (+0.032) | 0.607 (+0.017) | 0.596 (+0.006) |
| Task Arithmetic | 0.445 | 0.404 (−0.041) | 0.253 (−0.192) | 0.462 (+0.017) | 0.442 (−0.002) | 0.462 (+0.017) |
| TSV | 0.627 | 0.612 (−0.015) | 0.532 (−0.095) | 0.630 (+0.003) | 0.662 (+0.035) | 0.612 (−0.015) |
| TIES | 0.463 | 0.293 (−0.170) | 0.365 (−0.098) | 0.492 (+0.028) | 0.380 (−0.083) | 0.473 (+0.010) |
| DARE | 0.596 | 0.585 (−0.011) | 0.441 (−0.155) | 0.617 (+0.022) | 0.591 (−0.005) | 0.586 (−0.010) |
| **Avg $\Delta$** | — | −0.045 | **−0.149** | +0.020 | −0.008 | +0.002 |
| *ViT-B/32* | | | | | | |
| Weight Avg | 0.699 | 0.638 (−0.061) | 0.537 (−0.162) | 0.738 (+0.039) | 0.646 (−0.053) | 0.682 (−0.018) |
| Task Arithmetic | 0.619 | 0.632 (+0.013) | 0.522 (−0.097) | 0.613 (−0.006) | 0.608 (−0.011) | 0.483 (−0.136) |
| TSV | 0.692 | 0.637 (−0.055) | 0.511 (−0.181) | 0.688 (−0.004) | 0.684 (−0.008) | 0.688 (−0.004) |
| TIES | 0.543 | 0.539 (−0.004) | 0.492 (−0.050) | 0.554 (+0.012) | 0.547 (+0.004) | 0.436 (−0.107) |
| DARE | 0.684 | 0.723 (+0.038) | 0.519 (−0.165) | 0.707 (+0.022) | 0.689 (+0.004) | 0.654 (−0.031) |
| **Avg $\Delta$** | — | −0.014 | **−0.131** | +0.012 | −0.013 | −0.059 |

ods.

**Subspace Metrics Are Important but Architecture- and Method-Dependent.** Subspace metrics rank second in average importance ($\Delta r = -0.045$ on ViT-B/16, $-0.014$ on ViT-B/32). On ViT-B/16, the effect is concentrated in TIES ($\Delta r = -0.170$), while other methods are barely affected. On ViT-B/32, Weight Averaging ($-0.061$) and TSV ($-0.055$) suffer the most. The higher sensitivity of TSV to subspace metrics reflects its reliance on low-rank SVD compression of task vectors, where input-space subspace alignment is structurally relevant.

**Activation Metrics Are Architecture-Dependent.** On ViT-B/16, removing activation metrics causes negligible change ($\Delta r = +0.002$). On ViT-B/32, however, the drop is substantial ($\Delta r = -0.059$), driven by Task Arithmetic ($-0.136$) and TIES ($-0.107$). This suggests that activation-based signals encode architecture-specific information that becomes predictively relevant at smaller patch sizes (ViT-B/32 has fewer patches per image, making activation statistics more discriminative).

**Effective Rank, Task Vector, and Activation Metrics on ViT-B/16 Are Largely Non-Critical.** Removing effective rank metrics consistently *improves* validation performance ($+0.020$ on ViT-B/16, $+0.012$ on ViT-B/32), confirming that they introduce noise and are correctly suppressed by L1 regularization. Task vector metrics are marginal ($\Delta r \approx -0.01$ on both backbones), suggesting that the structural information captured by subspace and gradient metrics subsumes the simpler distance-based task vector signals.

### A.14.4. OBSERVATIONS

**Minimal Metric Set.** A reduced set containing only gradient-based and subspace metrics (12 total) captures most of the predictive power of the full 28-metric set, with gradient metrics being the indispensable component.

**Understanding Mergeability.** The dominance of gradient-based metrics suggests that mergeability is fundamentally about *functional compatibility*: models that respond to data with similar gradient landscapes require less destructive parameter averaging. Subspace metrics provide complementary structural information about which weight-space directions are modified. Together, they characterize both the functional and geometric aspects of task interference.

**Task Vector Metrics Are Largely Redundant.** Task vector metrics measure raw distances and angles in weight space. The ablation shows that this information is largely subsumed by the more structured gradient and subspace signals: once gradient compatibility and subspace alignment are accounted for, the raw magnitude and direction of task vectors add little additional predictive power.

## A.15. Hyperparameters

This appendix details all hyperparameters used in our experiments, including merging method configurations, metric computation settings, and optimization parameters. Let $M$ denote the number of tasks, which is 20 in this work.

### A.15.1. MERGING METHODS

Table 21 summarizes the hyperparameters for each merging method evaluated in our benchmark.

*Table 21.* Hyperparameters for each merging method.

| Method | Parameter | Value / Description |
|---|---|---|
| Task Arithmetic | $\alpha$ (scaling)
Formula | 0.3
$\theta_{\mathrm{merged}} = \theta_{\mathrm{pre}} + \alpha \sum_{t=1}^{M} \tau_t$ |
| Weight Averaging | $\alpha$ (scaling)
Formula | 1.0 (implicit, no scaling)
$\theta_{\mathrm{merged}} = \theta_{\mathrm{pre}} + \frac{1}{M} \sum_{t=1}^{M} \tau_t$ |
| TSV | SVD compression
Non-matrix aggregation | Per-task (compression ratio $= 1/M$)
Mean |
| TIES | $\alpha$ (scaling)
$k$ (trim keep ratio)
Sign resolution
Aggregation | 0.3
0.2 (retain top-20% by magnitude per task)
Mass-based majority vote
Disjoint mean over sign-agreeing entries |
| DARE | $\alpha$ (scaling)
Drop rate $p$
Rescaling
Aggregation | 1.0
0.9 (random Bernoulli drop on task-vector entries)
$1/(1-p)$ applied to surviving entries
Mean over masked, rescaled task vectors |

**Task Arithmetic.** We use a fixed scaling coefficient $\alpha = 0.3$, which has been found to work well across diverse task combinations in prior work (Ilharco et al., 2023).

**Weight Averaging.** It is a simple uniform average of the models and can alternatively be reduced to a special case of Task Arithmetic, where the scaling coefficient $\alpha$ is $\frac{1}{M}$.

**TSV (Task Singular Vectors).** The SVD compression ratio is set to $1/M$ where $M$ is the number of tasks being merged. Non-matrix parameters (e.g., biases, layer norms) are aggregated using the mean.

**TIES.** TIES Merging (Yadav et al., 2023) resolves interference between task vectors via three steps: (i) *Trim*: for each task, keep only the top-$k$ fraction of parameters by magnitude (we use $k = 0.2$, retaining the top 20%) and zero out the remainder; (ii) *Elect*: compute a per-coordinate consensus sign using mass-based majority voting, where the sign with the largest summed magnitude wins; (iii) *Disjoint Merge*: average only those trimmed entries whose sign agrees with the elected sign, ignoring the rest. The resulting consensus task vector is then added to the pretrained weights with scaling coefficient $\alpha = 0.3$.

**DARE.** DARE (Drop And REscale) (Yu et al., 2023) sparsifies each task vector independently before aggregation. For every task, each entry of the task vector is randomly dropped with probability $p = 0.9$ (Bernoulli mask), and the surviving entries are rescaled by $1/(1-p)$ to preserve the expected value. The masked, rescaled task vectors are then aggregated with their mean and added to the pretrained weights with scaling coefficient $\alpha = 1.0$. We use the random masking strategy with a fixed random seed for reproducibility.

### A.15.2. MERGEABILITY METRICS

Table 22 lists the hyperparameters used for computing mergeability metrics.

*Table 22.* Hyperparameters for mergeability metric computation.

| Category | Parameter | Value |
|---|---|---|
| Subspace Overlap | $k$ (top/bottom directions)
Singular value overlap $k$
Applied to
Layers | 10
100
Left and right singular vectors
All transformer blocks |
| Activation-Based | Calibration samples per task
Batch size
Random seed
Target layer | 10
32
42
`visual.transformer.resblocks.11` |
| Gradient-Based | Calibration samples per task
Batch size
Random seed
Gradient type | 10
8
42
Encoder and input gradients |

**Subspace Overlap Metrics.** For the left and right subspace overlap metrics, as well as the interaction matrix overlap, we use $k = 10$ singular directions from both the top (highest singular values) and bottom (lowest singular values) of the spectrum. This captures both the principal and residual subspaces of the task vectors. For singular value overlap, we use $k = 100$ to capture a broader distribution of the singular value spectrum.

**Activation-Based Metrics.** We extract activations from the last transformer block (`resblocks.11` for ViT-B-16) using 10 calibration samples per task. The activations are compared using $L_2$ distance, cosine similarity, magnitude ratio, and dot product.

**Gradient-Based Metrics.** We compute gradients with respect to both the encoder parameters and the input images. This requires a forward-backward pass on calibration data, using 10 samples per task with a batch size of 8 to manage memory constraints.

### A.15.3. LINEAR OPTIMIZATION

Table 23 details the hyperparameters for the learned linear mergeability predictor.

*Table 23.* Hyperparameters for linear mergeability optimization.

| Parameter | Value |
|---|---|
| Optimizer | Adam |
| Learning rate | 0.01 |
| Maximum iterations | 1,000 |
| Early stopping patience | 50 iterations |
| Convergence threshold | $10^{-4}$ |
| Metric normalization | Min-max to $[-1, 1]$ |
| Target metric | Normalized test accuracy (average) |
| Cross-validation | Leave-one-task-out (20 folds) |

### A.15.4. MLP

For the MLP ablation study (Appendix 7.2), we used the hyperparameters reported in Table 24.

### A.15.5. FINE-TUNING

The task-specific ViT models were fine-tuned from pretrained CLIP ViT-B/16 or B/32 checkpoints using the hyperparameters specified by the original authors, based on convergence characteristics. See Table 25.

*Table 24.* Hyperparameters for MLP mergeability predictor.

| Parameter | Value |
|---|---|
| Architecture | Input $\rightarrow$ Hidden $\rightarrow$ Output |
| Hidden dimension | 8 |
| Activation | ReLU |
| Dropout rate | 0.4 |
| Optimizer | Adam |
| Learning rate | 0.001 |
| Weight decay ($L_2$) | 0.001 |
| Epochs | 300 |
| Batch size | Full batch |
| Input normalization | Min-max to $[-1, 1]$ |

*Table 25.* Fine-tuning hyperparameters for ViT-B/16 vision models.

| Parameter | Value |
|---|---|
| Base model | CLIP ViT-B/16 (OpenAI) |
| Optimizer | AdamW |
| Batch size | 64 |
| Learning rate | $1 \times 10^{-5}$ |
| Weight decay | 0.1 |
| Gradient accumulation | 2 steps |
| Gradient clipping | 10.0 |
| Precision | FP32 |
| Epochs | Dataset-specific (1–147) |

The number of fine-tuning epochs follows the original authors of the checkpoints on Hugging Face. Epochs vary significantly across datasets, ranging from 1 epoch for PCAM to 147 epochs for Flowers102. Full values are shown in Table 26.

*Table 26.* Complete set of the 20 tasks and their number of fine-tuning epochs.

| Task | fine-tuning Epochs |
| --- | --- |
| Stanford Cars | 35 |
| CIFAR10 | 6 |
| CIFAR100 | 6 |
| DTD | 76 |
| EMNIST | 3 |
| EuroSAT | 12 |
| FashionMNIST | 5 |
| FER2013 | 10 |
| Flowers102 | 147 |
| Food101 | 4 |
| GTSRB | 11 |
| KMNIST | 5 |
| MNIST | 5 |
| OxfordIIITPet | 82 |
| PCAM | 1 |
| RenderedSST2 | 39 |
| RESISC45 | 15 |
| STL10 | 6 |
| SUN397 | 14 |
| SVHN | 4 |

## A.16. Datasets

Following Gargiulo et al. (2025), we run and evaluate our mergeability discovery framework on a benchmark of 20 diverse image classification tasks: SUN397 Xiao et al. (2016), Stanford Cars (Krause et al., 2013), RESISC45 (Cheng et al., 2017), EuroSAT (Helber et al., 2019), SVHN (Netzer et al., 2011), GTSRB (Stallkamp et al., 2011), MNIST (Lecun et al., 1998), DTD (Cimpoi et al., 2014), Flowers102 (Nilsback & Zisserman, 2008), PCAM (Veeling et al., 2018), FER2013 (Goodfellow et al., 2013), OxfordIIITPet (Parkhi et al., 2012), STL10 (Coates et al., 2011), CIFAR10 & CIFAR100 (Krizhevsky & Hinton, 2009), Food101 (Bossard et al., 2014), FashionMNIST (Xiao et al., 2017), EMNIST (Cohen et al., 2017), KMNIST (Clanuwat et al., 2018), and RenderedSST2 (Socher et al., 2013).

