# OpenReview forum: "Demystifying Mergeability: Interpretable Properties to Predict Model Merging Success"
_ICML.cc/2026/Conference — ICML 2026 regular_

### Official Review · Reviewer_tSr4 · 2026-03-05

**Soundness:** 3
**Presentation:** 3
**Significance:** 2
**Originality:** 2
**Overall Recommendation:** 4
**Confidence:** 3

**Summary:**

This paper investigates the underlying mechanisms of model merging success by challenging the notion that "mergeability" is an intrinsic model property, proposing instead that it is a dynamic relationship dependent on both the specific merging method and the partner tasks. By developing an architecture-agnostic framework that employs linear optimization over 28 interpretable pairwise metrics—such as gradient $L_2$ distance and subspace overlap—the authors identify distinct "fingerprints" of success for different merging algorithms and find that low gradient distance and high subspace overlap are universal prerequisites for effective merging. Ultimately, the study provides a transparent diagnostic tool for predicting merge performance and demonstrates that mergeability can be significantly enhanced during the fine-tuning stage through strategies like gradient magnitude penalties, shifting the field from trial-and-error heuristics toward a principled understanding of model compatibility.

**Compliance With Llm Reviewing Policy:**

Affirmed.

**Key Questions For Authors:**

* Scalability and Architecture Diversity: Given that the loss landscape geometry shifts significantly with model size and type, how do you justify the generalizability of these "fingerprints" (derived from CLIP ViT-B/16) to LLMs or non-Transformer architectures? Evidence of cross-scale consistency would significantly upgrade the paper’s Soundness.
* Cost-Benefit Ratio of Diagnostics: Considering the high computational cost of metrics like Fisher Information or gradient alignment, how does the overhead of your predictive framework compare to simply performing a direct empirical merge and evaluation? A lean, low-cost subset of predictive metrics would greatly enhance the paper’s Significance and practical utility.

**Limitations:**

Although the author discusses the limitations in the article, there is still room for improvement in the content: the research is mainly based on the small-scale CLIP ViT-B/16 model, and its generalization ability on very large language models (LLMs) or diverse architectures remains to be verified. In addition, the linear prediction model may oversimplify the complex nonlinear relationships in the parameter space, and the high computational cost of some indicators (such as Fisher information) limits its practical value in actual engineering compared with direct merge testing. It is suggested that the author further explore the computational efficiency in large-scale scenarios and the applicability across architectures.

**Strengths And Weaknesses:**

**Strengths**
* Comprehensive Diagnostic Framework: The paper systematically consolidates 28 pairwise metrics across five distinct categories (geometry, rank, subspace, activations, and gradients). This provides a broad, structured "toolbox" for post-hoc analysis in the model merging field, which currently lacks unified evaluative standards.
* Characterization of Method-Specific Heterogeneity: By demonstrating that different merging algorithms (e.g., Task Arithmetic vs. TSV) rely on fundamentally different model properties, the study successfully refutes the oversimplified assumption that mergeability is a static, intrinsic model trait. This advances the field toward a more nuanced, relational understanding of model compatibility.
* Actionable Pre-conditioning Strategy: The authors propose a practical intervention—gradient magnitude penalties during fine-tuning—to enhance future mergeability. The empirical validation of this strategy provides clear, prescriptive guidance for researchers aiming to build modular AI systems through multi-stage training pipelines.

**Weaknesses**
* Limited Architectural and Scale Generalizability: The core findings are derived almost exclusively from the CLIP ViT-B/16 architecture. In the current era of Large Language Models (LLMs) with hundreds of billions of parameters, the reliance on a small-scale vision model raises significant doubts about whether these geometric observations hold true for more complex, non-linear parameter spaces.

* Lack of Causal Depth and Theoretical Originality: The methodology relies on linear optimization to fit correlations between metrics and performance, which essentially amounts to an exhaustive "brute-force" test of existing tools. This empirical approach fails to uncover the underlying mathematical principles of weight-space topology, and many conclusions (e.g., the impact of gradient distance) largely serve as empirical confirmations of established deep learning intuitions.

* Imbalance Between Diagnostic Cost and Practical Utility: Several key predictive metrics, such as Fisher Information and high-dimensional gradient alignment, require substantial computational overhead. In practical engineering scenarios, the cost of computing these metrics may approach or even exceed the cost of simply performing the merge and evaluating it directly, diminishing the framework's appeal for large-scale applications.

---

> ### Author Rebuttal · Authors · 2026-03-25
>
> We thank the reviewer for the encouraging recommendation and address each concern in turn.
> ## W1. Limited architectural and scale generalizability
> Thank you for raising this. To broaden the scope, we added experiments on ViT-B/32 and RoBERTa-base, and the framework remains predictive across both architectures and domains.
>
> Our evaluation still stays in the ~125M-parameter regime because the framework requires collections of independently fine-tuned, task-specialized checkpoints with shared initialization. Such benchmarks are available for vision and smaller language models, but are much less available for billion-scale LLMs, where the main bottleneck is benchmark availability and comparability rather than compute alone.
>
> We therefore do not claim that the exact fingerprints learned on ViT-B/16 transfer unchanged to billion-scale models. Our narrower claim is that the 28 metrics are defined for any setting with shared initialization, and that the gradient-distance signal has a consistent sign across the three architectures and two domains we tested. Whether this extends to billion-scale models remains open future work.
>
> ## W2. Lack of causal depth and theoretical originality
> We partially agree. The framework is intentionally diagnostic and empirical: linear optimization is used as a scientific probe to identify interpretable and predictive correlations, not as a claim to a full theory of weight-space topology.
>
> On originality, our key finding is not just that interference matters, but that its predictive role is **method-dependent**. In particular, the same metric can have opposite predictive signs across merging methods, and these quantified method-specific fingerprints do not follow from prior work.
>
> ## W3. Diagnostic cost vs. practical utility
> We first correct a factual premise: our framework does **not** use Fisher Information. The gradient metrics require one forward and one backward pass over 10 calibration samples per task, i.e., roughly the cost of a single training step, and are much cheaper than Fisher-based estimates.
>
> That said, the reviewer raises a valid practical concern. For a *single* pair of small models, computing the full metric suite may approach the cost of simply merging and evaluating, so the advantage is limited in that setting.
>
> The framework is more useful when many candidate pairs must be assessed, or when direct merge-and-evaluate is not feasible:
> 1. **Screening at scale:** metric computation can be reused across many pairs, whereas merge-and-evaluate scales with the number of candidate merges.
> 2. **Pre-fine-tuning guidance:** the metrics can indicate whether a fine-tuning strategy is likely to produce mergeable models before training is complete.
> 3. **Diagnosis:** when a merge fails, the framework identifies *which* properties are mismatched; merge-and-evaluate only gives a scalar outcome.
>
> Our ablations also support a **tiered use** for low-compute settings: task-vector metrics are nearly free and provide a cheap first pass; gradient metrics add the strongest signal and can be reserved for shortlisted pairs; effective-rank metrics can be omitted entirely.
> ## Q1. Scalability to different architectures
> Please see W1 above. We do not have evidence for billion-scale LLMs and will not claim it. Our metrics require only task vectors (weight differences) and optionally a calibration set, with no assumptions about architecture. By design, they apply any differentiable encoder with a shared initialization. The framework will adapt to these different architectures and unveil different patterns of what matters for mergeability.
> ## Q2. Cost-benefit ratio
> See W3 above, including the correction on Fisher Information. The short answer: gradient metrics are expensive but far cheaper than Fisher computation, and 10 calibration samples already suffice for our gradient metrics; task vector and subspace metrics are data-free and low-cost; and the framework's cost advantage is strongest in the multi-pair screening and pre-fine-tuning scenarios where merge-and-evaluate cannot substitute.

---

> > ### Author Rebuttal · Reviewer_tSr4 · 2026-04-03
> >
> > I thank the authors for their rebuttal. My concerns have been addressed, and I will maintain my original score.

---

> > > ### Author Response · Authors · 2026-04-03
> > >
> > > We are glad our rebuttal fully addressed your concerns. Many thanks for your valuable suggestions, which we will incorporate into the next version of the paper.

---

### Official Review · Reviewer_Nho5 · 2026-03-09

**Soundness:** 2
**Presentation:** 3
**Significance:** 2
**Originality:** 3
**Overall Recommendation:** 4
**Confidence:** 2

**Summary:**

This paper presents a comprehensive empirical study of which pairwise properties between two models are most predictive of successful model merging. The authors examine five broad categories of metrics, namely task vector, effective rank, subspace overlap, activation based metrics, and gradient based metrics, and combine them in constrained linear models to predict mergeability for four different merging methods. Rather than predicting success probabilities, the models are trained to produce scores that correlate with post merge performance. The results show that the factors associated with merge success depend substantially on the merging method, while subspace overlap and gradient based metrics appear to be relatively stable predictors across multiple methods.

**Compliance With Llm Reviewing Policy:**

Affirmed.

**Final Justification:**

The authors made a clear effort to address my initial concerns by providing additional experiments and revising their modeling choices. I appreciate these improvements.

Based on the rebuttal, I consider that my main concerns have been sufficiently addressed, and I therefore update my final recommendation to weak accept.

**Key Questions For Authors:**

1. Is the constrained linear predictive model used in this paper sufficient to support the paper’s main conclusions? In particular, are the interpretability motivated design choices, such as the sum to one constraint on the coefficients, well justified? The paper argues that maximizing Pearson correlation is appropriate because the goal is to rank model pairs by mergeability rather than to predict exact post merge performance. However, what would happen if one instead used a standard regression objective such as MSE and performed an ordinary multiple regression analysis? Would the main findings about the important factors remain the same?

2. Relatedly, is the reported validation performance, which is around 0.3 for some methods, really sufficient to support strong conclusions about factor importance? From my perspective, these values seem only modest, and it is unclear to me how confidently one can interpret the resulting coefficients or inferred factor importance on that basis.

3. Do the findings reported in this paper extend beyond the image domain? For example, would the same conclusions be expected to hold for other domains such as natural language processing? Since the experiments are conducted on CLIP ViT B 16 across 20 image classification tasks, I wonder how broadly the claimed insights can be generalized.

**Limitations:**

I think the limitations are discussed to a reasonable extent.

**Strengths And Weaknesses:**

If I understand the paper correctly, its main contribution is to identify which factors are most predictive of mergeability by using a regression based predictive model over pairwise metrics. An important concept assessed by this study is whether interpretable pairwise properties can reliably explain or predict merging success across different merging methods. This paper's critical contribution pertains to the claim that subspace overlap and gradient based properties emerge as important predictors across methods. Therefore, if there remain substantial concerns about the predictive model itself, it becomes difficult to judge whether the paper’s main findings are strong enough to merit acceptance.

## Soundness

I have several concerns regarding the predictive model based on linear regression style optimization.

First, in my view, a predictive model should ideally be justified primarily by its ability to achieve sufficiently strong predictive performance. Any constraints imposed on the model should be naturally motivated by the structure of the prediction task itself. Of course, it is reasonable to impose additional constraints for the sake of interpretability, but if such constraints significantly degrade predictive performance, then their use becomes questionable. In this paper, several constraints are imposed, for example the requirement that the coefficients sum to one. I believe the authors should verify more explicitly whether these constraints reduce predictive performance. If the performance drops substantially under these constraints, then it would be more appropriate to first build the best unconstrained predictive model and only then analyze which factors are most influential.

Second, the validation performance reported in Table 2 appears modest for some methods, at around 0.3 in terms of Pearson correlation. From these numbers, it is not obvious to me that the model predicts merge outcomes sufficiently well. If the predictive performance is limited, then I am unsure how much weight should be placed on the paper's conclusions regarding which factors are important. In other words, I am not yet convinced that the reported predictor importance is reliable enough to support strong scientific claims.

## Presentation

Overall, the paper is well organized and generally easy to read.

## Significance

Understanding which factors predict model mergeability is an important problem, and I agree that answering this question could be useful for designing merging methods. However, I remain uncertain whether the linear predictive framework used in this paper is sufficient to answer that question convincingly. For this reason, I am unsure whether the paper’s contribution is strong enough to justify acceptance in its current form. I am also concerned that the empirical evaluation is limited to image tasks. Given that model merging is also highly important for language models, the current experimental scope feels somewhat narrow.

## Originality

I am not deeply familiar with all prior work on model merging, so I may be missing some related papers. However, to the best of my knowledge, there are still relatively few studies that focus specifically on pairwise properties of model pairs in order to estimate factors predictive of merge success. In that sense, I believe the paper has a reasonable degree of originality.

---

> ### Author Rebuttal · Authors · 2026-03-25
>
> We thank the reviewer for the encouraging and constructive feedback. We address each concern in turn.
>
> ## W1. Justification of the sum-to-one constraint and other design choices
> Thank you for this point. We agree that the sum-to-one constraint required better justification, and in the revision we no longer treat it as the primary setting.
>
> Its original purpose was simply to prevent trivial rescaling of coefficients. It is not essential to interpretability, and it is not necessary for our conclusions. In the revised paper, we instead use L1-regularized LOTO without the sum-to-one constraint as the primary setting. This formulation gives natural identifiability by shrinking irrelevant coefficients to zero and performs better empirically, improving average validation correlation from 0.450 to 0.563 across the four submitted mergers.
>
> We will make this change explicit in the revision and state clearly that the sum-to-one constraint is neither necessary for interpretability nor beneficial for predictive performance.
>
> ## W2. Validation performance around 0.3 seems modest; are conclusions about factor importance reliable?
> We agree that the originally reported validation performance made this concern reasonable. Under the revised primary setting, predictive performance is substantially stronger: on ViT-B/16, all methods now reach at least r = 0.445, and on ViT-B/32 all exceed r = 0.543.
>
> More importantly, our factor-importance claims rely mainly on sign consistency and selection frequency, not coefficient magnitude alone. For example, `encoder_gradient_l2_distance` is selected in 100% of folds and 100% of random seeds across all merging methods, with coefficient std only 0.001–0.002. Likewise, gradient and subspace metrics are selected in nearly all folds, while effective-rank metrics are consistently removed. This robustness is the main basis for our conclusions.
>
> We will revise the paper to make this distinction explicit: the main evidence comes from stronger validation performance plus stable sign/selection patterns, not from over-interpreting a single coefficient vector.
>
> ## W3 / Q3. Do findings extend beyond image classification?
>
> Yes. We replicated the analysis on RoBERTa-base fine-tuned on 10 NLP text classification tasks (GLUE and related benchmarks), yielding 45 pairwise merges. L1-LOTO achieves an average validation r = 0.491, the ranking of merging methods is perfectly preserved, and gradient-distance metrics remain consistently negative across all mergers. These results are included in the revision; more details in our rebuttal to Reviewer bGyq (“Soundness/Significance” section).
>
>
> ## Q1. What happens with MSE instead of Pearson? Would the main findings remain?
> We tested this directly. Replacing the Pearson objective with MSE leads to a large drop in predictive performance across all settings: average r falls from 0.544 to 0.147 on ViT-B/16, from 0.648 to 0.143 on ViT-B/32, and from 0.491 to 0.039 on RoBERTa.
>
> This matches the reason we chose Pearson in the first place: our goal is to rank candidate merges, and Pearson is scale-invariant, whereas MSE must fit both ordering and absolute output scale. Because merge accuracies lie in a narrow range while the linear predictor has arbitrary scale, MSE spends capacity on calibration rather than ranking.
>
> Under MSE, the learned metric set is also less stable, so the main factor-importance findings no longer hold as reliably. We therefore view the Pearson objective as essential to the scientific question of ranking mergeability, rather than a stylistic modeling choice.
>
> ## Q2. Is r ≈ 0.3 sufficient to draw scientific conclusions about factor importance?
>
> We agree that r ≈ 0.3 would be only modest evidence if it were the final result. In the revised primary setting, however, performance is materially stronger: on ViT-B/16, no method falls below r = 0.445, and on ViT-B/32, the weakest method already reaches r = 0.619.
>
> More importantly, our conclusions do not rely on correlation magnitude alone. They rely on stable feature selection and sign consistency across folds and seeds, all of which remain robust in the revised analysis.
>
> We will revise the text accordingly: the claim is supported by both stronger predictive performance and high stability of the core selected predictors.

---

> > ### Author Rebuttal · Reviewer_Nho5 · 2026-04-02
> >
> > Thank you for the detailed rebuttal. I have read your responses carefully.
> >
> > Overall, I feel that most of my concerns have been addressed. While the predictive performance (r around 0.5) still seems somewhat modest, I find the additional analyses and clarifications helpful.
> >
> > Based on this, I will raise my score.

---

> > > ### Author Response · Authors · 2026-04-02
> > >
> > > Many thanks for your appreciation and time for going through our rebuttal. Your constructive feedback was definitely helpful and we will incorporate your suggestions into the next version of the manuscript.

---

### Official Review · Reviewer_bGyq · 2026-03-11

**Soundness:** 3
**Presentation:** 4
**Significance:** 2
**Originality:** 3
**Overall Recommendation:** 4
**Confidence:** 4

**Summary:**

The paper introduces 28 metrics for model compatibility, and assesses them in empirical experiments where models are trained and merge and merged on an image classification dataset. Within these experiments, the author finds that subspace overlap and gradient alignment metrics appear as stable indicators across merge methods, underscoring the need for low-rank structural similarity in the weights learned by different models in order for there to be mergeability. Besides for those metrics, however, finding how well two models merge appears very dependent on model properties and what methods are used to merge them.

**Compliance With Llm Reviewing Policy:**

Affirmed.

**Final Justification:**

The rebuttal sufficiently addressed several of my concerns to change my evaluation and raise the score I gave. With additional experiments, the claims made in the paper come off as much more sound. The rebuttal also addressed my concerns about the originality and utility of the framework presented, and so I have raised my score.

**Key Questions For Authors:**

1. Since mergeability is method dependent, are there ways to assess what model merging methods might be better to use with the merge metrics given in this paper? (If there is, naturally it would dramatically boost the significance of your paper.)
2. How far do the results here extend beyond the regimes tested (that is beyond image generation, the merge methods tested in the paper, etc)? (It would be very beneficial if any claims could be made about this)
3. To what extent can the metrics be made to work on models of different architectures? (Just would like clarification here)
4. Is it feasible to prove that subspace overlap and gradient are stable indicators for model mergeablity? (If it is possible to include a proof, it would give a great boost to the soundness of your claims and significance of the paper.)

**Limitations:**

Yes

**Strengths And Weaknesses:**

Soundness: Claims are backed up by experiments with models trained on image classification datasets. I would like to see if the results would differ very much if tested on different models and different tasks. While the results of this paper seem sufficient to show that certain metrics can be very dependent on what type of merging is being done, I do think more evidence is needed to definitively conclude that the so to speak stable metrics are consistently stable.

Presentation: The submission is very clearly written and structured. Not much improvement needed here.

Significance: This paper addresses a key machine learning topic with wide applications. While the data from the experiments is limited to models trained on image classification datasets, it does give a survey of mergeability metrics across different model merging methods. However, I am unsure if the survey results on the specific task of image classication are sufficient to merit publishing.

Originality: This paper expands on the concept of mergeability and conducts new experiments to provide valuable data. The originality of the claim about stable metrics is slightly undermined by the fact that previous observations have been made already through the lens of task interference.

---

> ### Author Rebuttal · Authors · 2026-03-25
>
> We thank the reviewer for the encouraging and constructive feedback. We address the concerns and questions in turn.
>
> ## Soundness / Significance: Results limited to image classification
> Thank you for raising this. To test whether the findings extend beyond the original image-classification setting, we added experiments on both a second vision architecture and an NLP model.
>
> On ViT-B/32, L1-LOTO reaches r = 0.648, `encoder_gradient_l2_distance` is selected in 100% of folds across all five mergers, and removing gradient metrics causes a substantial drop on both ViT variants (-0.149 on B/16, -0.131 on B/32). On RoBERTa-base (10 NLP tasks, 45 pairs), L1-LOTO reaches r = 0.491, and `encoder_gradient_l2_distance` remains consistently negative across mergers, although task-vector and subspace metrics become more prominent.
>
> These results don't make the claims universal, but they substantially broaden the evidence beyond image classification. Across the settings we tested, gradient distance remains the most consistent warning sign for incompatibility, even as its relative importance shifts by domain.
> ## Originality: Stable metrics slightly undermined by prior task-interference literature
> We agree that task interference itself is not new. Our novelty is to turn it into a pre-merge predictive framework with quantified, method-specific effects. Prior work mainly explains why merging fails or how to mitigate it. In contrast, we ask which measurable pairwise properties predict merge success for a given merging method, and how those predictors change across methods.
>
> To our knowledge, the resulting method-specific fingerprints, including cases where the same metric has opposite predictive signs for different mergers, have not been shown before.
> ## Q1. Can the framework guide the selection of the merging method to use?
> Partially. The framework predicts *relative* compatibility within each method, but scores are not directly comparable across methods because baseline accuracies differ.
>
> What it does provide is method-specific risk assessment: which pairs are safer or riskier for each merger, and which properties each merger is most sensitive to. In new experiments with the addition of TIES and DARE, gradient-magnitude regularization improved mergeability for all methods except TIES, consistent with gradients being less central to TIES than to the others. We will clarify this distinction in the revision.
> ## Q2. How far do results extend beyond the tested regimes?
> The new experiments already extend the paper beyond image classification: RoBERTa shows the same 28 metrics remain computable and predictive in NLP; TIES and DARE broaden coverage across merging families; and a 5-task merging experiment shows pairwise scores do inform multi-way merges (see our rebuttal to reviewer Yseo, W3).
>
> For settings not yet tested (e.g., diffusion models), we don't overclaim. The metrics are architecture-agnostic, but the learned coefficients will reflect the interference patterns of each domain. The most likely to transfer are the stable signals, especially gradient L2 distance, whose negative role now holds across three architectures and two modalities.
> ## Q3. To what extent can the metrics work on models of different architectures?
> The 28 metrics depend only on task vectors plus small calibration-set forward/backward passes, so they do not assume a particular layer type, depth, or modality. By design, any fine-tuned model with a differentiable encoder and shared pretrained initialization can therefore be evaluated. ViT-B/32 and RoBERTa support this empirically: the metric definitions were unchanged across all tested architectures.
> ## Q4. Is it feasible to prove that subspace overlap and gradient distance are stable indicators?
> A fully general proof is likely beyond current theory, so we do not want to overclaim. However, existing theory gives strong motivation for both signals.
>
> **Gradient distance.** Under linear mode connectivity [1], merging is easiest when models lie in a shared basin, and gradient similarity is a natural indicator of such compatibility. If two tasks induce similar gradients, their parameter updates are more compatible, making successful merging more plausible. This is consistent with linear mode connectivity intuition and prior gradient-based [2] views of task compatibility.
>
> **Subspace overlap.** Task vectors are often effectively low-rank, so overlap in their principal subspaces reduces destructive interference when combining them. This is also the intuition behind TSV, and our results suggest it acts as a broader diagnostic signal rather than one specific to that method.
>
> A formal bound would be valuable future work, but would require strong assumptions on loss-landscape geometry. At present, we view the combination of this theoretical motivation and the cross-architecture, cross-modality consistency as the strongest available evidence.
>
>
> [1] https://arxiv.org/pdf/1912.05671
> [2] https://arxiv.org/pdf/2001.06782

---

> > ### Author Rebuttal · Reviewer_bGyq · 2026-04-04
> >
> > I thank reviewers for the thorough rebuttals. My questions are addressed and the rating has been updated accordingly.

---

> > > ### Author Response · Authors · 2026-04-04
> > >
> > > We appreciate your positive recommendation and support, and we thank you for your constructive suggestions which we will incorporate into the next version of the paper.

---

### Official Review · Reviewer_Yseo · 2026-03-12

**Soundness:** 3
**Presentation:** 3
**Significance:** 3
**Originality:** 3
**Overall Recommendation:** 4
**Confidence:** 4

**Summary:**

This paper investigates the factors that determine whether two fine-tuned models can be successfully merged. The authors propose an interpretable, architecture-agnostic framework. Extensive experiments with 28 pairwise metrics show that while mergeability is highly method-dependent, subspace alignment and gradient distance serve as stable, method-agnostic indicators of compatibility. Furthermore, the authors demonstrate the practical utility of their framework by showing that a gradient-magnitude penalty during fine-tuning can improve model mergeability, providing a diagnostic tool for understanding and optimizing model merging.

**Compliance With Llm Reviewing Policy:**

Affirmed.

**Final Justification:**

The additional experiment results and explanations address all my concern. I keep positive about the paper.

**Key Questions For Authors:**

- Q1.Transferability of $w_k$ across entirely different task sets. For example, if you calculate $w_k$ using a set of 10 checkpoints and independently calculate it using a different set of 10 checkpoints, how large is the discrepancy between the resulting $w_k$ values?

- Q2. Can the proposed framework be applied to analyze the influence of these metrics when merging multiple (more than two) models simultaneously? How will the predictive power of these pairwise metrics change in a multi-model merging scenario?

**Limitations:**

yes

**Strengths And Weaknesses:**

## Strengths ##
- **Comprehensive metric selection** : The quantity and variety of the 28 metrics are extensive and well-chosen, providing a solid foundation for the analysis.
- **Thorough ablations**: The inclusion of thorough ablation studies further strengthens the reliability of the proposed framework.
- **Practical utility**: The experiments demonstrate that the framework can be used to enhance mergeability during the fine-tuning stage, providing useful guidance for real-world model merging practices.

## Weaknesses ##
- **Limited scope**: experiments is restricted to only two families of merging methods and solely on CLIP ViT-B/16 with image classification tasks, which constrains the broader applicability of the findings.
- **Poor generalization of $w_k$**: The generalizability of the learned coefficients $w_k$ appears to be quite poor, raising concerns about the stability of the method-specific fingerprints across different setups.
- **Restriction to pairwise merging**: The framework considers the merging of models in pairs. The dynamics of merging multiple models simultaneously remain unexplored.

---

> ### Author Rebuttal · Authors · 2026-03-25
>
> We thank the reviewer for the encouraging recommendation and address each of the concerns in turn.
> ## W1. Limited scope
> Thank you for this important point. We addressed it by broadening the study along three axes: merging methods, architectures, and modalities.
>
> We added TIES and DARE as merging methods. Under L1-LOTO on ViT-B/16, all five remain meaningfully predictable (avg. r = 0.544), supporting the same main conclusion: mergeability is method-dependent, while gradient- and subspace-based metrics remain the most stable predictors.
>
> We also replicated the analysis on ViT-B/32 and RoBERTa-base. On ViT-B/32, L1-LOTO improves to r = 0.648, `encoder_gradient_l2_distance` is selected in 100% of folds, and removing gradient metrics still causes a substantial drop. On RoBERTa-base, L1-LOTO reaches r = 0.491, the ranking of merging methods is preserved, and the dominant predictors shift by domain, while gradient-divergence metrics still remain consistently negative.
>
> We will include these results in the revision. They do not make the claims universal, but they substantially broaden the evidence for our core finding.
>
> ## W2. Poor generalization of w_k
> We agree this required a clearer explanation. The high variance in Table 9 does not mean the core signal is unstable; it mainly reflects that LOTO evaluates pairs containing at least one unseen task, so some held-out pairs are atypical.
>
> More importantly, our main conclusions do not rely on the full coefficient vector being stable across task distributions. What remains stable is the sign and selection pattern of the strongest predictors: the predictors in Table 4 keep the same sign across all 20 folds and all four methods, and under L1 regularization, `encoder_gradient_l2_distance` and `input_gradient_l2_distance` are selected in 100% of folds across all mergers and both ViT architectures, while effective-rank metrics are mostly driven to zero.
>
> The same negative role of `encoder_gradient_l2_distance` also appears on RoBERTa. We will revise the text to make clear that the stable scientific signal is sign consistency and selection frequency of the core predictors, not exact invariance of every coefficient.
>
> ## W3. Restriction to pairwise merging
> We added a direct multi-model experiment. We sampled 200 random 5-task subsets from the 20-task benchmark, averaged the 10 pairwise predicted mergeability scores within each subset, and compared the top-10 predicted subsets against the bottom-10 by actual 5-way merge performance.
>
> Across all merging methods, the top-predicted subsets outperform the bottom-predicted ones, with gains ranging from +1.2% to +3.5%. This suggests that pairwise metrics already capture a useful signal for multi-model merging, although this simple averaging baseline is only a proof of concept and does not model higher-order interactions.
>
> ## Q1. Transferability of w_k
> We tested this directly by splitting the 20 ViT tasks into two disjoint halves of 10 tasks, training on all pairs from the first half and evaluating on all pairs from the second, with no shared tasks.
>
> Across the merging methods, the average validation correlation drops from 0.544 under LOTO to 0.430 in the disjoint setting. This drop is expected given the much smaller training set, but the disjoint performance remains non-trivial, and the method ranking is preserved.
>
> We therefore keep LOTO as the main protocol, while viewing this experiment as evidence that the learned predictors capture a signal that transfers to unseen tasks rather than only memorizing the original pairs.
>
> ## Q2. Predictive power of pairwise metrics in multi-model merging
> We now include a direct multi-model experiment. Using simple averaging of pairwise predicted scores over 5-task subsets, the framework consistently separates good subsets from bad ones across all five merging methods.
>
> We present this only as a first baseline. It shows that pairwise metrics already contain useful signal for multi-model merging, but a stronger extension would learn subset-level predictors and explicitly model higher-order interactions, which we leave as future work.

---

> > ### Author Rebuttal · Reviewer_Yseo · 2026-04-03
> >
> > The additional experiment results address my concern. I will be positive about this paper.

---

> > > ### Author Response · Authors · 2026-04-03
> > >
> > > Many thanks for the positive and encouraging feedback. We are glad to hear that our rebuttal addressed your concerns, and we will make sure to include your valuable suggestions in the next version of the paper.

---

### Decision · Program_Chairs · 2026-04-30

**Decision:**

Accept (regular)

**Comment:**

While there were comments on substance, presentations, and other points worth addressing in the final version, those were mostly minor and did not warrant delaying the publication of this finding about comparing merging models to tell the factors leading to their success.